# Integrative proteomic and lipidomic analysis of *GNB1* and *SCARB2* knockdown in human subcutaneous adipocytes

**Takuya Kitamoto**◉°, **Aya Kitamoto**◉°*

Advanced Research Facilities and Services, Division of Preeminent Research Supports, Institute of Photonics Medicine, Hamamatsu University School of Medicine, Hamamatsu, Shizuoka, Japan

◉ These authors contributed equally to this work.
* a.ktmt@hama-med.ac.jp

## Abstract

Obesity, a global public health concern, is influenced by various factors, including genetic predispositions. Although many obesity-associated genes have been identified through genome-wide association studies (GWAS), the molecular mechanisms linking these genes to adipose tissue function remain largely unexplored. This study integrates proteomic data on adipocyte fat accumulation with GWAS data on obesity to unravel the roles of the identified key candidate genes — G protein subunit beta 1 (*GNB1*) and scavenger receptor class B member 2 (*SCARB2*) — involved in fat accumulation. We utilized RNA interference to knock down *GNB1* and *SCARB2* in human subcutaneous adipocytes, followed by lipidome and proteome analyses using mass spectrometry. Knockdown of these genes resulted in a reduction in lipid droplet accumulation, indicating their role in adipocyte lipid storage. Digital PCR confirmed effective gene knockdown, with *GNB1* and *SCARB2* mRNA levels significantly reduced. In total, the lipidomic analysis identified 96 lipid species with significant alterations. *GNB1* knockdown resulted in a decrease in cholesterol esters and an increase in phosphatidylcholines, phosphatidylinositols, and ceramides. *SCARB2* knockdown also led to an increase in phosphatidylcholines, with a trend towards decreased triacylglycerols. Proteomic analysis revealed significant changes in proteins involved in lipid metabolism and adipocyte function, including PLPP1 and CDH13, which were upregulated following *GNB1* knockdown, and HSPA8, which was downregulated. Conversely, *SCARB2* knockdown resulted in the downregulation of PLPP1 and METTL7A, and the upregulation of PLIN2, HSPA8, NPC2, and SQSTM1. Our findings highlight the significant roles of *GNB1* and *SCARB2* in lipid metabolism and adipocyte function, providing insights that could inform therapeutic strategies targeting these regulatory genes in obesity.

## Introduction

Obesity is a global public health issue characterized by excessive body fat accumulation [1]. It is influenced by diet, physical activity, and genetic and environmental factors. Recent studies have highlighted the significant role of genetic variants in obesity, leading to the extensive use of

**Data availability statement:** The data underlying this study are publicly available at the Metabolomics Workbench (https://www.metabolomicsworkbench.org) with Study ID: ST003452 (DOI: http://dx.doi.org/10.21228/M8T52T).

**Funding:** This work was supported by the Japan Society for the Promotion of Science KAKENHI (grant number: JP23K10957).

**Competing interests:** The authors have declared that no competing interests exist.

genome-wide association studies (GWAS) to identify obesity-related genes [2–17]. Subcutaneous adipose tissue, located beneath the skin, plays a vital role in energy storage and metabolism and contributes to overall metabolic health [18–20]; however, its biology remains elusive. Moreover, adipose tissue is intricately involved in the development of obesity through various mechanisms. Nevertheless, despite the discovery of many obesity-associated genes, the molecular mechanisms linking these genes to adipose tissue function remain largely unexplored. To address this gap, integrating various "omics" data that have traditionally been considered in isolation has gained increased interest. We hypothesized that integrating proteomic data on adipocyte fat accumulation with GWAS data on obesity would provide new insights into the roles of obesity-related genes in fat accumulation. Previously, we conducted a comprehensive proteomic analysis of human subcutaneous white adipocytes during differentiation and maturation using mass spectrometry (MS). This analysis identified several proteins that varied significantly during adipocyte differentiation and fat accumulation [21]. We then integrated these findings with body mass index (BMI)-related GWAS data from 82 studies, encompassing numerous genes, conducted in Japan and internationally up to 2015 [17]. This integration allowed us to screen and identify candidate genes. Among these, we focused on genes whose protein levels showed an overall increase throughout the differentiation and maturation of human subcutaneous white adipocytes, as evidenced by our comprehensive proteomic analysis. Specifically, we selected G protein subunit beta 1 (*GNB1*) and scavenger receptor class B member 2 (*SCARB2*) for further investigation, as their proteins exhibited remarkable and continuous expression changes during adipocyte development. As illustrated in S1 Fig, which presents graphs generated using numerical data from our previous study [21], both GNB1 and SCARB2 proteins showed increased expression during adipocyte differentiation and fat accumulation compared to the pre-differentiation stage. This trend suggests their potential roles in these processes.

*GNB1* encodes the beta subunit of heterotrimeric guanine nucleotide-binding proteins (G proteins), which are essential components functioning directly downstream of G protein-coupled receptors (GPCRs). Upon ligand binding to the GPCR, the G protein (Gαβγ) is activated and dissociates into Gα and Gβγ subunits. These dissociated subunits then activate downstream signaling enzymes, with Gα specifically regulating cellular levels of inositol trisphosphate ($IP_3$) and diacylglycerol (DG; Fig 1A) [22]. This G protein-mediated signaling cascade is critical for virtually all cells in mammals and may play a key role in adipocyte function and metabolism [22]. While the specific role of *GNB1* in adipocyte metabolism remains to be fully elucidated, some studies have suggested a potential involvement in the regulation of lipolysis [23].

The major lysosomal membrane proteins (LMPs) identified to date include LAMP1 and LAMP2 [24], as well as SCARB2 (also known as LIMP-2) [25, 26]. SCARB2, a highly glycosylated type III membrane protein, is predominantly located in the membranes of late endosomes and lysosomes (Fig 1B) [27, 28]. SCARB2 plays a crucial role in lysosomal function and may be involved in lipid metabolism and storage [29].

In this study, we employed RNA interference to knockdown the candidate genes *GNB1* and *SCARB2* in human subcutaneous adipocytes, followed by lipidome and proteome analyses using MS. This approach enabled us to identify specific lipid and protein species that were altered upon gene knockdown. Our findings hold promise for developing therapeutic strategies targeting these regulatory genes in subcutaneous adipocytes.

## Materials and methods

### Cell culture

The primary cultured human preadipocytes used for the experiment were obtained from Zen-Bio, Inc. (Research Triangle Park, NC, USA). This cell line of human subcutaneous preadipocytes

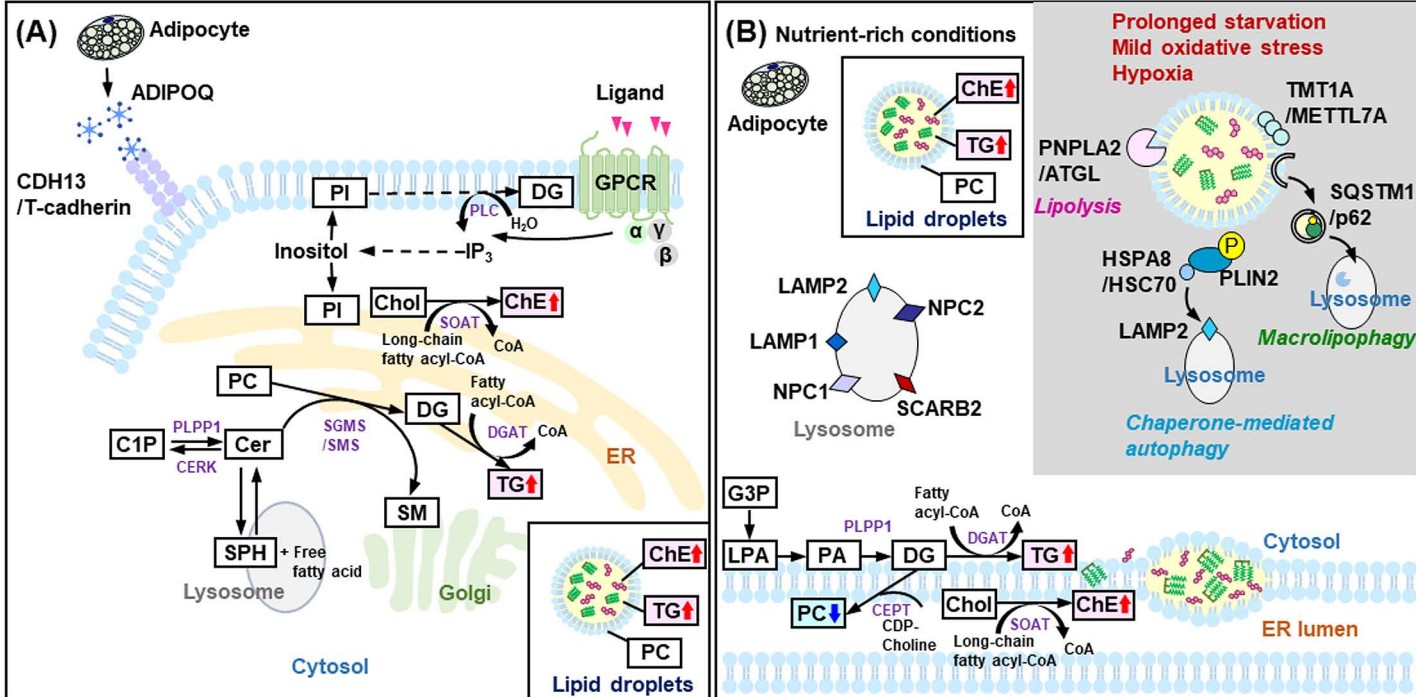

**Fig 1. GNB1 and SCARB2 involvement in lipid metabolism of human subcutaneous adipocytes.** (A) A schematic representation of the lipid biosynthesis and signaling pathways related to G protein subunit beta 1 (GNB1) in human subcutaneous adipocytes. Under nutrient-rich conditions, lipid droplets accumulate in the cytoplasm. The core of the lipid droplets is rich in neutral lipids, mainly triglycerides (TG), and cholesteryl esters (ChE). (B) The state of human subcutaneous adipocytes before the knockdown of *SCARB2* is shown. Lipid droplets accumulate in the cytoplasm under nutrient-rich conditions. In contrast, during prolonged starvation, mild oxidative stress, or hypoxia, lipid droplets are degraded through processes such as lipolysis, chaperone-mediated autophagy, or macrolipophagy. In the schematic diagram, significant increases are indicated by red arrows pointing upwards, while significant decreases are shown by blue arrows pointing downwards. Abbreviations: ADIPOQ, adiponectin, C1Q and collagen domain containing; α, alpha subunit of the G proteins; β, GNB1; C1P, ceramide-1-phosphate; Cer, ceramide; CERK, ceramide kinase; CDH13/T-cadherin, cadherin 13; CEPT, choline/ethanolaminephosphotransferase; ChE, cholesterol ester; Chol, cholesterol; DG, diacylglycerol; DGAT, diacylglycerol O-acyltransferase; ER, endoplasmic reticulum; G3P, glycero-3-phosphate; GPCR, G protein-coupled receptor; γ, gamma subunit of the G proteins; HSPA8/HSC70, heat shock protein family A; IP$_3$, inositol (1,4,5)-trisphosphate; LAMP1/2, lysosomal associated membrane protein 1/2; LPA, lysophosphatidic acid; NPC1/2, NPC intracellular cholesterol transporter 1/2; PA, phosphatidic acid; PC, phosphatidylcholine; PI, phosphatidylinositol; PLIN2, perilipin 2; PLPP1, phospholipid phosphatase 1; PLC, phospholipase C; PNPLA2/ATGL, patatin like phospholipase domain containing 2; SCARB2, scavenger receptor class B member 2; SGMS/SMS, sphingomyelin synthase; SM, sphingomyelin; SOAT, sterol O-acyltransferase; SPH, sphingosine; SQSTM1/p62, sequestosome 1; TG, triacylglycerol; TMT1A/METTL7A, methyltransferase-like protein 7A.

was derived from the abdominal subcutaneous adipose tissue of a 38-year-old Caucasian woman. Upon arrival, the preadipocytes were thawed in a 37 °C water bath with agitation and then transferred to a centrifuge tube containing 10 mL Preadipocyte Medium (#PM-1, Zen-Bio, Inc.). The cells were centrifuged at 280 × *g* for 5 min at 20 °C. After centrifugation, the supernatant was aspirated, and the cell pellet was resuspended in fresh Preadipocyte Medium.

To obtain the necessary number of cells for the experiment, human subcutaneous white preadipocytes were initially seeded into 90 mm diameter cell culture dishes (#20100, SPL Life Sciences Co., Ltd., Korea) and cultured at 37 °C with 5% CO₂ in a humidified incubator using Preadipocyte Medium containing DMEM/Ham's F-12 (1:1, v/v), 2-[4-(2-hydroxyethyl)-1-piperazinyl] ethanesulfonic acid (HEPES) (pH 7.4), fetal bovine serum (FBS), penicillin, streptomycin, amphotericin B, and 3.15 g/L D-glucose. After reaching 80% confluence, the medium was aspirated from the cultured preadipocytes, and cells were washed four times using HEPES-buffered balanced salt solution and treated with trypsin/EDTA (0.25%/1x) solution (Gibco, Grand Island, NY, USA) at 37 °C. The trypsinization was neutralized with fresh Preadipocyte Medium, and the cells were subsequently counted using a hemocytometer. The counted human subcutaneous white preadipocytes

were plated onto a 12-well plate (#30012, SPL Life Sciences Co., Ltd., Korea) at a density of 8,000 cells/cm² and cultured at 37 °C with 5% $CO_2$ in a humidified incubator in Preadipocyte Medium until 100% confluence was reached (the day that the preadipocytes achieved 100% confluence was designated as day 0). Subsequently, differentiation into human subcutaneous white adipocytes was induced using Adipocyte Differentiation Medium (#DM-2, Zen-Bio, Inc.) comprising DMEM/Ham's F-12 (1:1, v/v), HEPES (pH 7.4), FBS, biotin, pantothenate, human insulin, dexamethasone, 3-isobutyl-1-methylxanthine, peroxisome proliferator-activated receptor gamma agonist, penicillin, streptomycin, amphotericin B, and 3.15 g/L D-glucose, for four consecutive days.

After an additional 72 h of incubation in Adipocyte Differentiation Medium, the medium was replaced with Adipocyte Maintenance Medium (#AM-1, Zen-Bio, Inc.) comprising DMEM/Ham's F-12 (1:1, v/v), HEPES (pH 7.4), FBS, biotin, pantothenate, human insulin, dexamethasone, penicillin, streptomycin, amphotericin B, and 3.15 g/L D-glucose. Following another 72 h of incubation in Adipocyte Maintenance Medium, the medium was replaced with fresh Adipocyte Maintenance Medium, and the culture was continued for an additional 96 h. During this culture period, small lipid droplets began to appear in the cytoplasm, and by day 14, the cytoplasm was filled with numerous lipid droplets, indicating the maturation of the human subcutaneous white adipocytes. All experiments were performed in three independent replicates (Fig 2).

Human subcutaneous white preadipocytes were seeded in a 12-well plate, and the day they reached 100% confluence in Preadipocyte Medium was designated as day 0. The preadipocytes were then differentiated for four consecutive days in Adipocyte Differentiation Medium. Four days after initiating differentiation, the adipocytes underwent the first transfection with siRNA. After 72 h of the first transfection, the medium was replaced with Adipocyte Maintenance Medium, and the adipocytes were then subjected to a second transfection. After 72 h of the second transfection, the adipocytes underwent a third transfection. The adipocytes, having been transfected three times, were cultured for 96 h and extracted for mRNA expression, lipidomic, and proteomic analysis using the respective application methods.

## siRNA transfection in human subcutaneous adipocytes

After differentiating human subcutaneous white preadipocytes for four consecutive days in Adipocyte Differentiation Medium, we performed the first of three siRNA transfections using MISSION® siRNAs (pre-designed siRNA) targeting human *GNB1* (#SASI_Hs01_0024621, Sigma-Aldrich, Saint Louis, MO, USA) and human *SCARB2* (#SASI_Hs01_0022831, Sigma-Aldrich). MISSION® siRNA Universal Negative Control #1 (#SIC001, Sigma-Aldrich) was used as the control. Each siRNA was diluted in serum-free Basal Medium (#BM-1, Zen-Bio, Inc.), comprising DMEM/Ham's F-12 (1:1, v/v), HEPES (pH 7.4), biotin, pantothenate, and 3.15 g/L D-glucose, followed by vortexing. INTERFERin siRNA transfection reagent (Polyplus, Illkirch, France) was then added. After vortexing for 10 s, the mixture was incubated at room temperature (20–25 °C) for 10 min to form the siRNA-INTERFERin complex. Subsequently, the medium was replaced with 2 mL fresh Adipocyte Differentiation Medium, to which 200 μL siRNA-INTERFERin complex was gently added and mixed to introduce each siRNA into the differentiated adipocytes. The final concentration of each siRNA was 50 nM.

After 72 h, a second siRNA transfection was performed. Briefly, 1.4 mL Adipocyte Differentiation Medium was removed from the first transfected adipocytes and replaced with 1.6 mL fresh Adipocyte Maintenance Medium. Subsequently, 200 μL siRNA-INTERFERin complex was added and gently mixed. After an additional 72 h, a third siRNA transfection was performed. Specifically, 1.4 mL of Adipocyte Maintenance Medium was replaced with 1.2 mL of fresh Adipocyte Maintenance Medium, to which 200 μL siRNA-INTERFERin complex was added and gently mixed. For all transfections, the final concentration of each siRNA was maintained at 50 nM.

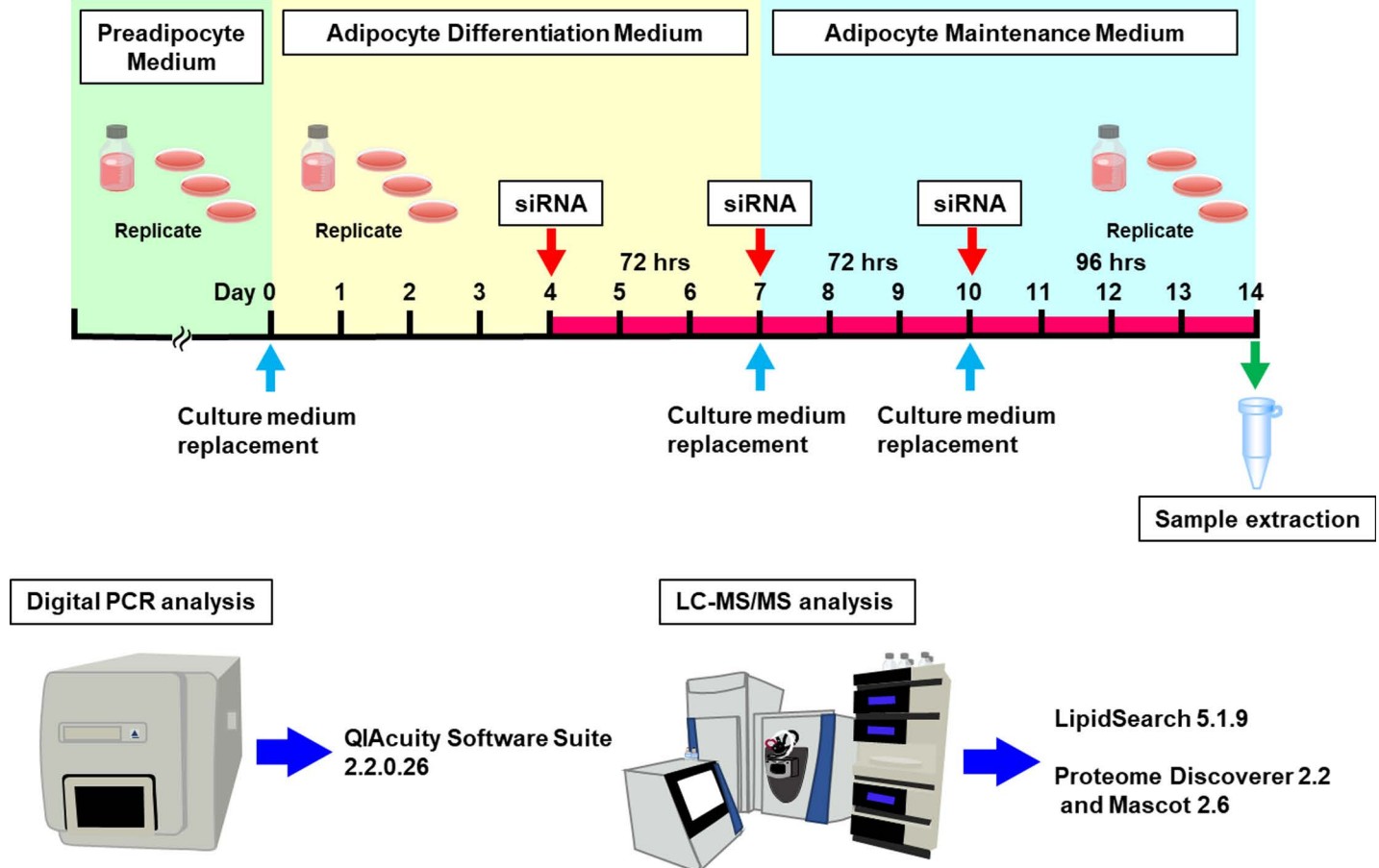

**Fig 2. Schematic representation of the experimental procedure: cell culture to mRNA expression, lipidomic, and proteomic analysis.**

The adipocytes, which had been transfected three times, were cultured for 96 h, i.e., 14 days after induction of differentiation with Adipocyte Differentiation Medium, and the cells were then washed twice with 1X PBS buffer. Subsequently, the cells were flooded with 200 µL 1X PBS buffer and gently scraped with a cell scraper; this was repeated three times, and the cells were transferred to 1.5 mL tubes. The collected cells were then centrifuged at 9,300 × g for 5 min, the supernatant was discarded, and the cell pellet was suspended in 200 µL 1X PBS buffer. This washing process was repeated twice, and the final pellet was suspended in 300 µL Milli-Q water and frozen at − 80 °C as a suspension cell sample until further use. Transfection efficiency was assessed using digital PCR (dPCR).

## Image acquisition

Phase contrast images of the human subcutaneous adipocytes transfected with siRNA-negative control, siRNA-*GNB1*, and siRNA-*SCARB2* were captured using cellSens Dimension software version 1.12, equipped with an Olympus IX83 inverted microscope (Olympus, Tokyo, Japan) and a 10X objective lens. Phase contrast image analysis was performed using ImageJ Fiji Software (NIH, Bethesda, MD, USA) [30–33] and the GNU Image Manipulation Program 2.10.38 (GIMP: https://www.gimp.org/).

## RNA extraction, cDNA synthesis, and primer design

Total RNA was extracted from the frozen suspension cell samples of human subcutaneous adipocytes using an RNeasy Lipid Tissue Mini Kit (Qiagen, Hilden, Germany) and eluted in 60 µL RNase-free water. Extraction blanks were processed alongside the experimental samples under identical conditions. The concentration and quality of the total RNA were assessed using an Agilent 4150 TapeStation system with an Agilent High Sensitivity RNA kit (Agilent Technologies, Santa Clara, CA, USA), with 1 µL total RNA used for analysis. The RNA integrity number (RIN) was > 5 for six samples and > 4 for the remaining three. Extracted total RNA was stored at − 80 °C. The total RNA samples were reverse-transcribed into cDNA using a ReverTra Ace® qPCR RT Master Mix with gDNA Remover (TOYOBO, Osaka, Japan), using 100 ng concentrated total RNA. The cDNA was either stored at − 20 °C or analyzed immediately using dPCR. The dPCR primers (S1 Table) were designed to avoid both forward and reverse primers being located within the same exon and to minimize the presence of single nucleotide polymorphisms on the primers. Additionally, the dPCR primers were confirmed to be unique across the human genome sequence (NCBI GRCh37) using Bowtie 2 (Version 2.3.5, https://bowtie-bio.sourceforge.net/bowtie2/index.shtml) [34].

## Absolute quantification of *GNB1* and *SCARB2* knockdown by digital PCR and statistical analysis

Gene expression was measured using the QIAcuity One digital PCR System with the QIAcuity EG PCR Kit (Qiagen, Hilden, Germany). The dPCR analysis program details are provided in S2 Table. The reaction mixtures comprised 10 ng target gene cDNA, 5 ng endogenous control gene cDNA, 1 × EvaGreen PCR Master Mix (Qiagen), and 0.4 µM of each primer. The final reaction volume of 12 µL was loaded onto a QIAcuity Nanoplate 8.5k 24-well (Qiagen), which was then sealed with the QIAcuity Nanoplate Seal (Qiagen). The thermal cycling conditions were set as follows: initial heat activation at 95 °C for 2 min, followed by 40 cycles of denaturation at 95 °C for 15 s, annealing at 60 °C for 15 s, and extension at 72 °C for 15 s. A final cooling cycle was performed at 40 °C for 5 min. Fluorescence data were collected using the green channel, and the concentration was automatically calculated using QIAcuity Software Suite version 2.2.0.26 (Qiagen). The images were exposed for 210 ms at a gain of 6, automatic settings were used for threshold and baseline adjustments. To enhance the accuracy of concentration measurements, variations between wells and batches in the nanoplate were adjusted by applying the Volume Precision Factor (VPF), as recommended by the manufacturer. Absolute quantification was calculated using Poisson statistics, and the final concentration of cDNA was expressed in unit copies/µL. The expression data of the target genes were normalized to that of Actin beta (*ACTB*), the endogenous control gene, using the following formula:

$$Normalization = \frac{Mean\ \left(copies\ /\ \mu L\right)\ of\ the\ target\ gene}{Mean\ \left(copies\ /\ \mu L\right)\ of\ the\ endogenous\ control\ gene\ \left(ACTB\right) \times 2}$$

In the above formula, the coefficient for *ACTB* was set to 2, reflecting that the concentration of *ACTB* is half that of the target gene. dPCR experiments were conducted following the Minimum Information for Publication of Quantitative Digital PCR Experiments (dMIQE) guidelines (S3 Table) [35, 36]. The dPCR data were compared between the siRNA-negative control and the siRNA-target genes using Student's *t*-test, with a significance level set at a *P*-value of < 0.05. Statistical analyses were performed using R software (http://www.r-project.org/).

## Analysis of adipogenic marker expression

To evaluate the impact of gene knockdown on adipocyte differentiation, we analyzed the expression levels of established adipogenic markers. These included the master regulators of adipogenesis peroxisome proliferator activated receptor gamma (*PPARG*) and CCAAT enhancer binding protein alpha (*CEBPA*), which encode essential transcription factors that coordinate the adipogenic program [37], and adipocyte-specific genes, fatty acid binding protein 4 (*FABP4*) [38] and adiponectin, C1Q and collagen domain containing (*ADIPOQ*) [39], which are reliable markers of mature adipocytes.

Total RNA extraction and cDNA synthesis were performed as described above. Quantitative real-time PCR (qPCR) was conducted using a QuantStudio 3 Real-Time PCR System (Applied Biosystems, USA) with THUNDERBIRD® NextSYBR™ qPCR Mix (#QPX-201, TOYOBO, Osaka, Japan). As with dPCR, the primer sequences used for qPCR are listed in S1 Table. To select an appropriate reference gene according to the Minimum Information for Publication of Quantitative Real-Time PCR Experiments (MIQE) guidelines [40], we evaluated *ACTB* and glyceraldehyde-3-phosphate dehydrogenase (*GAPDH*) expression in subcutaneous preadipocytes and mature subcutaneous adipocytes. For both cell types, the RIN values were > 7. Using the same RNA extraction and cDNA synthesis methods described above, qPCR analysis was performed using 10 ng of cDNA synthesized from the total RNA extracted from both cell types. *ACTB* showed consistent expression levels between preadipocytes and mature adipocytes (quantification cycle (Cq) = 19.16 (0.04) and 19.34 (0.02), respectively), while *GAPDH* expression varied due to its involvement in differentiation (Cq = 21.60 (0.01) and 20.96 (0.01), respectively). Based on these results, *ACTB* was selected as the reference gene due to its stable expression across different differentiation stages. No amplification was observed in the no-template controls (NTC). *ACTB* exhibited small variation (difference between means = 0.17 Cq), while *GAPDH* showed large differences between preadipocytes and mature adipocytes (difference between means = 0.63 Cq). The substantially smaller mean difference for *ACTB* supported its selection as a more suitable reference gene for normalization. Based on the stability of its expression between cell types, *ACTB* was selected as the reference gene for subsequent analyses.

Following the selection of *ACTB* as the reference gene, we assessed the PCR efficiency of each primer set. The Cq values were obtained from technical triplicates of serially diluted cDNA samples (10.0, 7.5, 5.0, and 2.5 ng). The slope, Y-intercept, and PCR efficiency were automatically calculated by QuantStudio™ Design and Analysis Software v1.5.2 (Applied Biosystems) based on the standard curve. PCR efficiency was determined using the equation: $E (\%) = (10^{-1/slope} - 1) \times 100$. The linearity of the standard curve was evaluated by the correlation coefficient ($R^2$). The PCR efficiencies and correlation coefficients for each primer set were as follows: *ACTB* (E = 91.8%, $R^2$ = 0.996, slope = −3.536); target genes: *PPARG* (E = 99.5%, $R^2$ = 0.996, slope = −3.334), *CEBPA* (E = 88.2%, $R^2$ = 0.971, slope = −3.641), *FABP4* (E = 95.6%, $R^2$ = 1, slope = −3.433), and *ADIPOQ* (E = 94.7%, $R^2$ = 0.995, slope = −3.457). The thermal cycling conditions were as follows: initial denaturation at 95 °C for 30 s, followed by 40 cycles of denaturation at 95 °C for 5 s and annealing/extension at 60 °C for 30 s. Melting curve analysis was performed to confirm the specificity of amplification. Gene expression was normalized to *ACTB* using the $2^{-\Delta\Delta Cq}$ method. The PCR products were evaluated for amplicon size using agarose gel electrophoresis and for specificity using melting curve analysis (data not shown). Statistical analysis was performed using one-way analysis of variance (ANOVA) followed by false discovery rate (FDR) correction for multiple testing using R software.

## Lipid extraction and tandem mass spectrometry (LC-MS/MS)

From the protein concentration determined using the bicinchoninic acid (BCA) assay, a volume corresponding to 5 µg of protein was prepared and adjusted to 200 µL with water. The sample was spiked with 5 µL of Splash Lipidomix (Avanti, Alabaster, AL, USA) as the internal standard. Subsequently, lipid extraction was performed using the Bligh and Dyer method [41], following the protocol detailed in our previous study [42]. The extracted lipids were reconstituted in 30 µL LC-MS-grade methanol, and 8 µL of the final solution was analyzed using a high-resolution mass spectrometer (Q-Exactive Hybrid Quadrupole-Orbitrap, Thermo Fisher Scientific, Waltham, MA, USA) coupled to a high-performance liquid chromatography system (Ultimate 3000, Thermo Fisher Scientific). The conditions for LC and MS were as previously described [42].

## Lipidomic analysis

Lipid species were identified using LipidSearch™ software version 5.1.9 (Thermo Fisher Scientific) with the following search conditions: the mass tolerance was configured to 5 ppm for precursor ions and 8 ppm for product ions, with the rank parameter set to 1. The database was configured to "Default (mammalian lipids (plasma, tissue, cells))." For alignment, the retention time (RT) tolerance was set to 1 min, RT Correction Tolerance to 0.5 min, Intensity Ratio Threshold to 1.5, and Valid Peak Rate Threshold to 0.5. A manual filter condition of "(S/N) > 10" was applied. The following selections were made based on the aligned results. Main Ion = M + H for lipid classes phosphatidylcholine (PC), lysophosphatidylcholine (LPC), ceramide (Cer), and sphingomyelin (SM); Main Ion = M + NH$_4$ for diacylglycerol (DG), triacylglycerol (TG), cholesterol ester (ChE), and phosphatidylglycerol (PG); Main Ion = M + H or M – H for phosphatidylethanolamine (PE), lysophosphatidylethanolamine (LPE), and phosphatidylserine (PS); Main Ion = M + NH$_4$ or M – H for phosphatidylinositol (PI); Main Ion = M – H for phosphatidic acid (PA); Grades A, B, and C for PC, SM, and sphingosine (SPH), otherwise, Grades A and B [43].

The filtered results were exported to an Excel file, presented in S4 Table. Area values for each lipid class were normalized using the Splash Lipidomix Standard added before sample preparation. A CSV file containing lipid species and normalized area values was created (S5 Table) and analyzed using MetaboAnalyst 6.0 (https://www.metaboanalyst.ca/) [44]. The data were log-transformed and autoscaled. Dendrogram, heatmap, partial least squares discriminant analysis (PLS-DA), and ANOVA were performed using MetaboAnalyst 6.0. Lipid data were analyzed using one-way ANOVA followed by FDR correction. Pairwise comparisons were performed using Fisher's least significant difference (LSD) post hoc tests on species showing significance after FDR correction ($P < 0.05$).

## Protein extraction and LC-MS/MS

A 60 µL frozen suspension cell sample was lysed by adding 240 µL lysis buffer (Mammalian Cell PE LB; G-Biosciences, St. Louis, MO, USA). The mixture was gently mixed at room temperature for 10 min. Protein concentration was determined using the BCA assay employing a Pierce Micro BCA Protein Assay Kit (Pierce, Rockford, IL, USA). Five micrograms of protein sample was precipitated by incubation in five volumes of ice-cold acetone at – 20 °C for 2 h. The precipitate was collected by centrifugation at 13,000 × *g* for 15 min at 4 °C, after which the supernatant was carefully aspirated. The resulting protein pellet was desiccated using a miVac Duo HV Plus (Genevac, Ipswich, England). Protein reduction, alkylation, and trypsin digestion were performed as previously described [21], with modifications to the amount of protein and trypsin used. Briefly, the precipitate was resuspended

in 25.5 μL 50 mM ammonium bicarbonate (FUJIFILM Wako Pure Chemical, Osaka, Japan), and 1.5 μL 500 mM dithiothreitol (FUJIFILM Wako Pure Chemical) was added. The mixture was heated at 95 °C for 5 min, followed by cooling to room temperature. Subsequently, 3 μL 500 mM iodoacetamide (FUJIFILM Wako Pure Chemical) was added, and the solution was incubated in darkness at room temperature for 20 min. For proteolytic digestion, 1 μL of trypsin solution (100 ng/μL) was added to the sample, which was then incubated at 37 °C for 4 h. An additional 1 μL trypsin (100 ng/μL) was added, and digestion was allowed to proceed overnight at 30 °C. The reaction was terminated by the addition of trifluoroacetic acid (Kanto Kagaku, Tokyo, Japan) to a final concentration of 5%. Peptides were purified using a MonoSpin C18 column (GL Science, Tokyo, Japan). The purified peptides were dried in a miVac Duo HV Plus and reconstituted in 30 μL of 0.1% formic acid (Kanto Kagaku). Finally, 8 μL of the peptide solution was analyzed using an EASY-nLC 1200 and a Q-Exactive Orbitrap mass spectrometer (Thermo Fisher Scientific). The LC and MS conditions were as described in a previous study [21].

### Proteomic analysis

Protein identification and relative quantification of the raw MS files were conducted using Proteome Discoverer 2.2 (Thermo Fisher Scientific) as described previously [21]. The Swiss-Prot human database (Version: 2023_01) was searched using the Mascot 2.6 search engine. The following search parameters were applied: fragment mass tolerance, 0.02 Da; precursor mass tolerance, 10 ppm; dynamic modifications, methionine oxidation; static modification, cysteine carbamidomethylation; enzyme name, trypsin; maximum missed cleavage sites, 2; and peptide confidence, high.

To facilitate the comparison of protein amounts across samples, the total peptide amount was normalized. The results obtained from Proteome Discoverer 2.2 were exported to an Excel file (S6 Table). From this output, proteins identified with high confidence were selected, and those with a relative standard deviation of the normalized abundance value exceeding 30% for any of the negative control, *GNB1*, or *SCARB2* groups were excluded. For further analysis in MetaboAnalyst 6.0, a CSV file was created containing the names and normalized abundance values of these filtered proteins (provided in S7 Table). Using MetaboAnalyst 6.0, the following analyses were performed: Proteins with missing normalized abundance values exceeding 70% were excluded. The remaining missing values were replaced with 1/5 of the minimum positive value of each variable. The data were then log-transformed and autoscaled. Subsequently, dendrogram, heatmap, and PLS-DA were performed. Finally, protein data were analyzed using one-way ANOVA followed by FDR correction. Pairwise comparisons were performed using Fisher's LSD post hoc tests on proteins showing significance after FDR correction ($P < 0.05$). All mass spectrometry data from this study were deposited to Metabolomics Workbench [45] for public access.

## Results

### Effects of siRNA knockdown on lipid droplets in human subcutaneous adipocytes

Three rounds of siRNA transfections targeting each gene (*GNB1* and *SCARB2*) were performed on human subcutaneous adipocytes. To visualize the experimental outcomes after knockdown of the target genes, phase contrast images were captured (Fig 3). The siRNA-negative control adipocytes exhibited numerous lipid droplets in the cytoplasm, some of which were slightly enlarged. In contrast, adipocytes with *GNB1* and *SCARB2* knockdown displayed fewer lipid droplets in the cytoplasm compared to the siRNA-negative control (Fig 3).

Each image corresponds to an independent replicate, as all experiments were conducted three times separately. The observed decrease in lipid droplets indicates the effectiveness of the target gene knockdown. Phase contrast images of human subcutaneous adipocytes transfected with siRNA-negative control, siRNA-G protein subunit beta 1 (*GNB1*), and siRNA-scavenger receptor class B member 2 (*SCARB2*). Scale bars: 100 μm.

The absolute quantification of mRNA levels of the target genes in human subcutaneous adipocytes following knockdown using siRNA-negative control, siRNA-*GNB1*, and siRNA-*SCARB2*, are shown in S2 Fig. The number of cDNA copies of *GNB1* was 191.3 (64.1) copies/μL after knockdown with siRNA-negative control but decreased to 27.4 (3.5) copies/μL after knockdown with siRNA-*GNB1*. Similarly, the number of cDNA copies of *SCARB2* reduced from 93.3 (20.5) copies/μL (with siRNA-negative control knockdown) to 44.7 (5.5) copies/μL (with siRNA-*SCARB2* knockdown; Table 1). The numbers represent means, and the parentheses show standard deviations. All dPCR data related to the knockdown experiments are summarized in S8 Table.

To provide a comprehensive assessment of the differentiation process, we analyzed the expression levels of adipogenic master regulator genes (*PPARG* and *CEBPA*) and adipocyte-specific genes (*FABP4* and *ADIPOQ*) using qPCR. In *GNB1* knockdown adipocytes, no significant differences were observed in the expression levels of any of the four genes when compared to that of the control group. Similarly, in *SCARB2* knockdown adipocytes, none of the genes showed significant changes compared to the control group. The relative expression levels of these genes are shown in S3 Fig, and their detailed qPCR data, including the p-values and FDR-adjusted values, are provided in S9 Table.

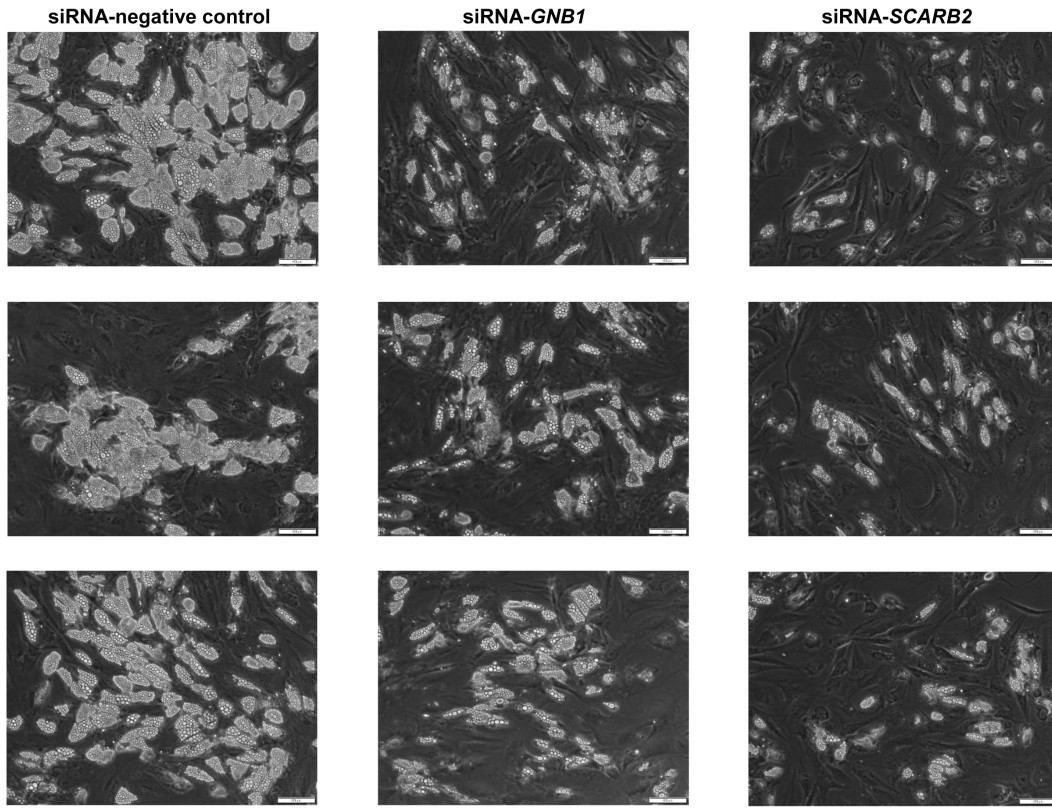

**Fig 3. Image analysis of three independent replicate experiments of subcutaneous adipocytes with target gene knockdown.**

**Table 1. Comparison of target gene knockdown levels using digital PCR.**

| Gene | Concentration (copies/μL) | | Normalization | | *P*-value |
|---|---|---|---|---|---|
| | siRNA-nc | siRNA-target gene | siRNA-nc | siRNA-target gene | |
| *GNB1* | 191.3 (64.1) | 27.4 (3.5) | 0.078 (0.031) | 0.020 (0.0016) | 0.030 |
| *ACTB* | 1244.5 (247.6) | 690.7 (441.1) | | | |
| *SCARB2* | 93.3 (20.5) | 44.7 (5.5) | 0.038 (0.0099) | 0.012 (0.0018) | 0.011 |
| *ACTB* | 1244.5 (247.6) | 1898.9 (306.4) | | | |

Data are presented as the mean (SD). Normalization was calculated as described in the methods section. *P*-values for the comparison of gene knockdown levels between the two groups were determined using Student's *t*-tests with normalized values. Abbreviations: *GNB1,* G protein subunit beta 1; *SCARB2,* scavenger receptor class B member 2; *ACTB,* actin beta; siRNA-nc, siRNA-negative control; SD, standard deviation.

Fig 4 illustrates the relative expression levels of *GNB1* and *SCARB2* genes following siRNA knockdown, as measured by digital PCR. These results demonstrate the effective knockdown of both *GNB1* and *SCARB2* genes using their respective siRNAs.

The absolute quantification of mRNA levels in human subcutaneous adipocytes was performed following knockdown using siRNA-negative control, siRNA-*GNB1*, and siRNA-*SCARB2*. Bars represent means (SD). The comparison of gene knockdown levels between the two groups was assessed using a one-tailed Student's *t*-test with normalized values. Black bars indicate siRNA-negative control and white bars indicate siRNA-target genes. Abbreviations: G, siRNA-G protein subunit beta 1 (*GNB1*); S, siRNA-scavenger receptor class B member 2 (*SCARB2*); SD, standard deviation.

## Alteration of lipid composition by targeted siRNA treatment

Processing of the raw data using LipidSearch 5.1.9 identified 366 lipid species (S4 Table). The dendrogram and PLS-DA of the lipid species separated the knockdown cells into distinct groups (Fig 5A, B).

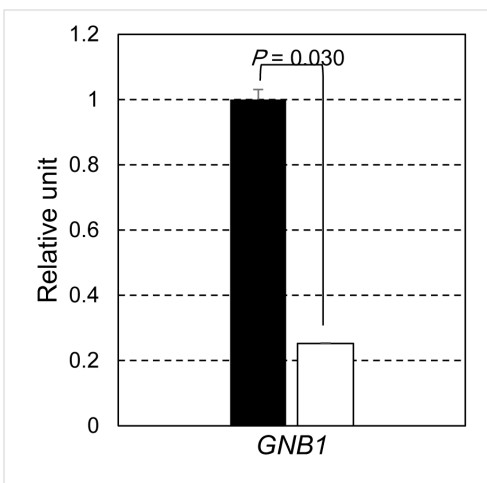
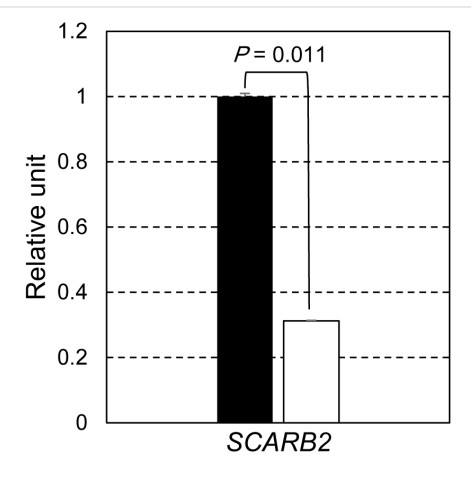
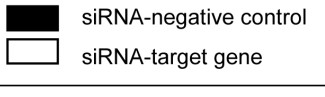

**Fig 4. Comparison of mRNA expression of *GNB1* and *SCARB2* using digital PCR.**

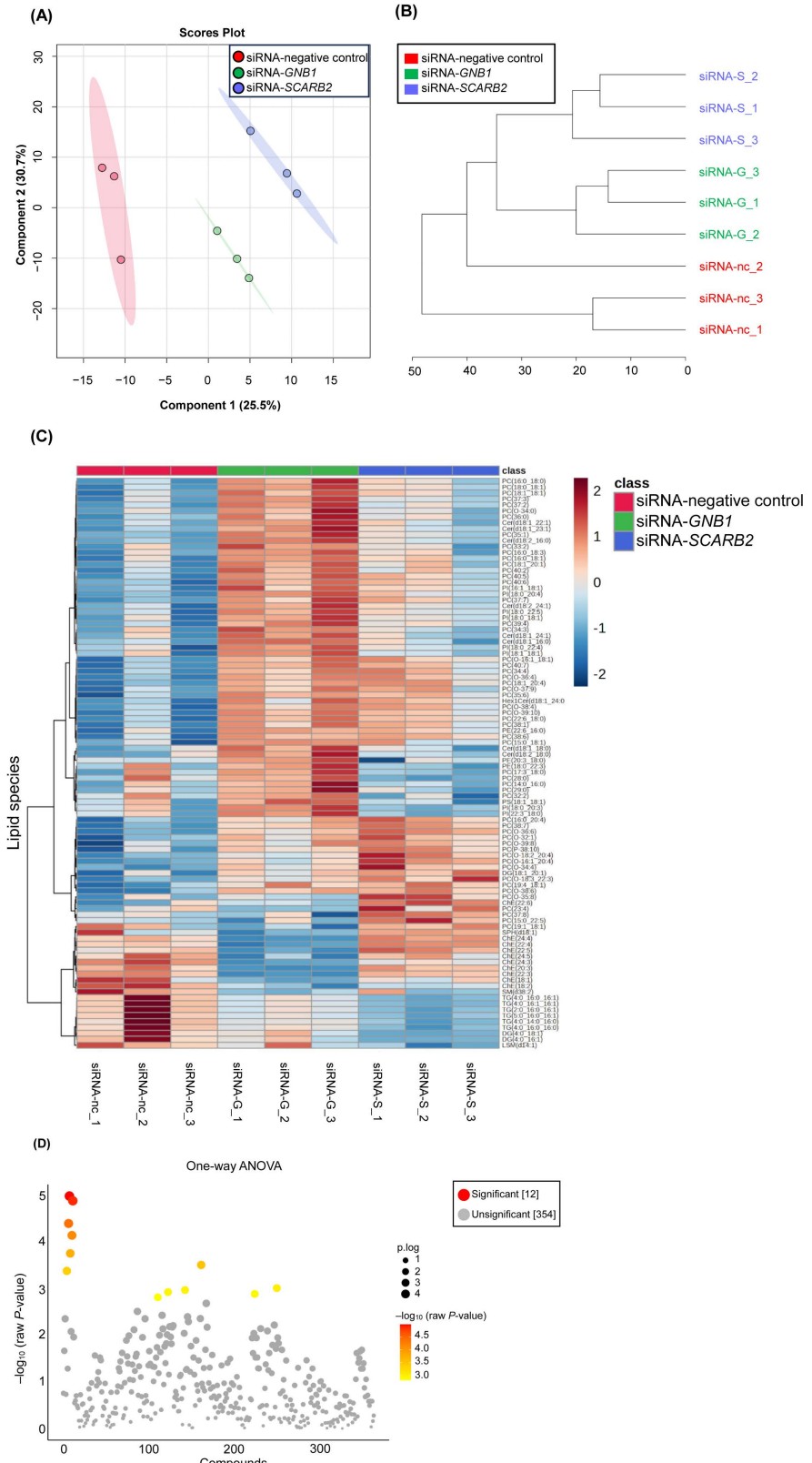

**Fig 5. Differential lipid species analysis in siRNA-transfected human subcutaneous adipocytes.** (A) Partial Least Squares Discriminant Analysis (PLS-DA) of lipids in the siRNA-transfected human subcutaneous adipocytes.

(B) Dendrogram showing the relationship between lipid species in the siRNA-transfected human subcutaneous adipocytes using Euclidean distances and Ward's clustering. (C) Hierarchical clustering heatmap analysis in the siRNA-transfected human subcutaneous adipocytes was performed using Ward's method with the Euclidean distance. Heatmaps show the top 96 lipid species that differed in the siRNA-transfected human subcutaneous adipocytes. The number at the bottom indicates the replicate number. The color scale indicates the number of standard deviations from the overall average of the lipid species, with areas in dark red indicating a higher amount of the lipid species, whereas areas in dark blue indicate a lower amount of the lipid species. The red, green, and blue colors represent cells transfected with the siRNA-negative control, siRNA-*GNB1*, and siRNA-*SCARB2*, respectively. (D) Scatterplots of *P*-values were obtained using one-way analysis of variance (ANOVA) of the 366 lipid species in the siRNA-transfected human subcutaneous adipocytes. The intensity of the red dots corresponds to the $-\log_{10}$ transformed *P*-values obtained using one-way ANOVA ($-\log_{10}$ (raw *P*-value)). After FDR correction of the ANOVA results, Fisher's LSD post hoc tests were performed on significant lipids to determine specific group differences. Both figures were generated using MetaboAnalyst 6.0. Abbreviations: siRNA-nc, siRNA-negative control; siRNA-G, siRNA-G protein subunit beta 1 (*GNB1*); siRNA-S, siRNA-scavenger receptor class B member 2 (*SCARB2*).

Statistical analysis of the data from 366 lipid species using one-way ANOVA identified 96 lipid species showing significant differences (critical value $F_{0.05}$ (2, 6) = 5.14, $P < 0.05$; Table 2). After FDR correction, 12 lipid species remained significant (adjusted $P < 0.05$). Pairwise comparisons using Fisher's LSD post hoc tests were performed on all species showing significant differences in ANOVA to determine specific group differences (Table 2).

Heatmap analysis, conducted using MetaboAnalyst 6.0, illustrated the results for 96 lipid species that showed significant differences (ANOVA, $P < 0.05$; Fig 5C). In adipocytes with *GNB1* knockdown, nine ChEs were significantly decreased compared to those with negative control knockdown (ANOVA, $P < 0.05$), with six of these remaining significant after FDR correction (adjusted $P < 0.05$). Fisher's LSD post hoc tests were then performed to identify specific group differences. Conversely, 44 PCs, eight Cers, and eight PIs were significantly increased (ANOVA, $P < 0.05$) with four PCs, one Cer, and one PI remaining significant after FDR correction (adjusted $P < 0.05$). In adipocytes with *SCARB2* knockdown, 28 PCs and one PI increased significantly compared to that in negative control knockdown adipocytes (ANOVA, $P < 0.05$). Among these, three PCs and one PI remained significant after FDR correction (adjusted $P < 0.05$). In contrast to *GNB1* knockdown, no significant changes were observed in Cers, but six TGs were significantly decreased (ANOVA, $P < 0.05$). A summary of the lipid species affected by each knockdown is provided in Table 3.

Detailed information on the lipid species with significant differences is presented in Table 2. Box and whisker plots for the 12 lipid species that remained significant after FDR correction (adjusted $P < 0.05$) are shown in S4 Fig. The scatterplots illustrating the *P*-values from one-way ANOVA are shown in Fig 5D. In these scatterplots, smaller *P*-values obtained from one-way ANOVA (indicating a greater significant difference) are represented by larger dots (p.log) and colors transitioning from yellow to red ($-\log_{10}$ (raw*P*-value)).

## Alteration of protein composition by targeted siRNA treatment

Processing of the raw data using Proteome Discoverer 2.2 identified 1,875 proteins, of which 1,316 proteins were retained after relative standard deviation value- and missing data-based filtration. The dendrogram and PLS-DA of the proteins separated the knockdown cells into distinct groups (Fig 6A, B).

Statistical analysis using one-way ANOVA on the data from 1,316 proteins identified 499 proteins that showed significant differences (critical value $F_{0.05}$ (2, 6) = 5.14, $P < 0.05$). After FDR correction, 141 proteins still showed significant alterations in at least one of the knockdown conditions (adjusted $P < 0.05$). Subsequent pairwise comparisons using Fisher's LSD post hoc tests on these proteins identified 45 proteins that were significantly increased and 48

**Table 2. Summary of statistically significant lipid species.**

| Lipid species | F-value | P-value | $-\log_{10}$ (P-value) | FDR adjusted P-value | Pairwise comparisons |
|---|---|---|---|---|---|
| ChE(22:4) | 135.04 | $1.03 \times 10^{-5}$ | 4.99 | 0.0024 | nc-G; S-nc; S-G |
| ChE(24:4) | 124.78 | $1.29 \times 10^{-5}$ | 4.89 | 0.0024 | nc-G; S-nc; S-G |
| ChE(22:3) | 84.96 | $3.97 \times 10^{-5}$ | 4.40 | 0.0048 | nc-G; S-G |
| ChE(24:3) | 69.33 | $7.14 \times 10^{-5}$ | 4.15 | 0.0065 | nc-G; nc-S; S-G |
| ChE(22:5) | 50.80 | $1.73 \times 10^{-4}$ | 3.76 | 0.013 | nc-G; S-nc; S-G |
| PC(O-36:6) | 41.52 | $3.06 \times 10^{-4}$ | 3.51 | 0.019 | G-nc; S-nc; S-G |
| ChE(20:3) | 37.36 | $4.11 \times 10^{-4}$ | 3.39 | 0.021 | nc-G; S-G |
| Cer(d18:2_16:0) | 27.42 | $9.60 \times 10^{-4}$ | 3.02 | 0.043 | G-nc; G-S |
| PC(O-16:1_20:4) | 26.45 | 0.0011 | 2.98 | 0.043 | G-nc; S-nc; S-G |
| PC(38:7) | 25.49 | 0.0012 | 2.93 | 0.043 | G-nc; S-nc |
| PI(18:0_20:4) | 24.63 | 0.0013 | 2.89 | 0.043 | G-nc; S-nc; G-S |
| PC(35:1) | 23.16 | 0.0015 | 2.82 | 0.046 | G-nc; G-S |
| PC(O-38:6) | 20.67 | 0.0020 | 2.69 | 0.057 | G-nc; S-nc |
| PC(18:1_20:4) | 17.74 | 0.0030 | 2.52 | 0.079 | G-nc; S-nc |
| PC(37:2) | 16.45 | 0.0037 | 2.44 | 0.084 | G-nc; G-S |
| PC(O-18:2_20:4) | 15.98 | 0.0039 | 2.40 | 0.084 | G-nc; S-nc; S-G |
| PC(O-36:4) | 15.44 | 0.0043 | 2.37 | 0.084 | G-nc; S-nc |
| PC(22:6_18:0) | 15.38 | 0.0043 | 2.36 | 0.084 | G-nc; S-nc |
| ChE(18:2) | 15.37 | 0.0044 | 2.36 | 0.084 | nc-G; nc-S |
| PC(O-18:3_22:3) | 14.53 | 0.0050 | 2.30 | 0.092 | G-nc; S-nc |
| Cer(d18:1_23:1) | 13.59 | 0.0059 | 2.23 | 0.096 | G-nc; G-S |
| PC(40:7) | 13.42 | 0.0061 | 2.21 | 0.096 | G-nc; S-nc |
| PC(39:4) | 13.16 | 0.0064 | 2.19 | 0.096 | G-nc; G-S |
| PC(40:6) | 13.04 | 0.0065 | 2.18 | 0.096 | G-nc; S-nc |
| PC(O-39:10) | 13.00 | 0.0066 | 2.18 | 0.096 | G-nc; S-nc |
| PI(22:3_18:0) | 12.64 | 0.0071 | 2.15 | 0.097 | G-nc; G-S |
| PC(37:3) | 12.59 | 0.0071 | 2.15 | 0.097 | G-nc; G-S |
| PC(18:0_18:1) | 12.35 | 0.0075 | 2.13 | 0.098 | G-nc; G-S |
| Cer(d18:1_16:0) | 12.06 | 0.0079 | 2.10 | 0.099 | G-nc; G-S |
| ChE(22:6) | 11.92 | 0.0081 | 2.09 | 0.099 | S-nc; S-G |
| PC(O-16:1_18:1) | 11.49 | 0.0089 | 2.05 | 0.10 | G-nc; S-nc |
| PI(18:0_20:3) | 11.48 | 0.0089 | 2.05 | 0.10 | G-nc; G-S |
| PC(38:6) | 11.16 | 0.0095 | 2.02 | 0.10 | G-nc; S-nc |
| PC(35:6) | 11.03 | 0.0098 | 2.01 | 0.10 | G-nc; S-nc |
| PC(16:0_20:4) | 10.92 | 0.010 | 2.00 | 0.10 | G-nc; S-nc |
| PC(40:2) | 10.65 | 0.011 | 1.97 | 0.10 | G-nc |
| ChE(24:5) | 10.61 | 0.011 | 1.97 | 0.10 | nc-G; S-G |
| PC(19:4_18:1) | 10.54 | 0.011 | 1.96 | 0.10 | G-nc; S-nc |
| Cer(d18:2_18:0) | 10.49 | 0.011 | 1.96 | 0.10 | G-nc; G-S |
| PI(18:0_22:5) | 10.43 | 0.011 | 1.95 | 0.10 | G-nc; G-S |
| PI(16:1_18:1) | 10.40 | 0.011 | 1.95 | 0.10 | G-nc |
| PC(16:0_18:0) | 9.88 | 0.013 | 1.90 | 0.11 | G-nc; G-S |
| PC(O-34:4) | 9.88 | 0.013 | 1.90 | 0.11 | G-nc; S-nc |
| Cer(d18:2_24:1) | 9.27 | 0.015 | 1.83 | 0.12 | G-nc |
| PC(38:1) | 9.26 | 0.015 | 1.83 | 0.12 | G-nc; S-nc |
| PC(40:5) | 9.18 | 0.015 | 1.83 | 0.12 | G-nc; S-nc |

*(Continued)*

**Table 2.** (Continued)

| Lipid species | *F*-value | *P*-value | $-\log_{10}$ (*P*-value) | FDR adjusted *P*-value | Pairwise comparisons |
|---|---|---|---|---|---|
| Cer(d18:1_22:1) | 8.94 | 0.016 | 1.80 | 0.12 | G-nc; G-S |
| PC(O-32:1) | 8.73 | 0.017 | 1.78 | 0.13 | G-nc; S-nc |
| PC(37:7) | 8.66 | 0.017 | 1.77 | 0.13 | G-nc; S-nc |
| PC(34:4) | 8.49 | 0.018 | 1.75 | 0.13 | G-nc; S-nc |
| Cer(d18:1_24:1) | 8.26 | 0.019 | 1.72 | 0.14 | G-nc; G-S |
| PC(O-38:4) | 8.05 | 0.020 | 1.70 | 0.14 | G-nc; S-nc |
| TG(4:0_16:1_16:1) | 8.03 | 0.020 | 1.70 | 0.14 | nc-S |
| Hex1Cer(d18:1_24:0) | 8.03 | 0.020 | 1.70 | 0.14 | G-nc; S-nc |
| ChE(18:1) | 7.85 | 0.021 | 1.67 | 0.14 | nc-G |
| PC(28:0) | 7.79 | 0.022 | 1.67 | 0.14 | nc-S; G-S |
| TG(4:0_16:0_16:1) | 7.77 | 0.022 | 1.67 | 0.14 | nc-S |
| PC(36:0) | 7.63 | 0.022 | 1.65 | 0.14 | G-nc; G-S |
| PC(15:0_22:5) | 7.50 | 0.023 | 1.63 | 0.14 | S-nc; S-G |
| Cer(d18:1_18:0) | 7.47 | 0.024 | 1.63 | 0.14 | G-nc; G-S |
| TG(2:0_16:0_16:1) | 7.47 | 0.024 | 1.63 | 0.14 | nc-S |
| PI(18:0_18:1) | 7.45 | 0.024 | 1.63 | 0.14 | G-nc |
| PC(18:1_20:1) | 7.32 | 0.025 | 1.61 | 0.14 | G-nc |
| PC(P-38:10) | 7.14 | 0.026 | 1.59 | 0.15 | G-nc; S-nc |
| DG(18:1_20:1) | 7.06 | 0.027 | 1.58 | 0.15 | G-nc; S-nc |
| PC(16:0_18:1) | 6.90 | 0.028 | 1.56 | 0.15 | G-nc |
| PC(18:1_18:1) | 6.88 | 0.028 | 1.55 | 0.15 | G-nc |
| PI(18:1_18:1) | 6.73 | 0.029 | 1.53 | 0.15 | G-nc; G-S |
| PC(O-34:0) | 6.72 | 0.029 | 1.53 | 0.15 | G-nc; G-S |
| PI(18:0_22:4) | 6.72 | 0.029 | 1.53 | 0.15 | G-nc; G-S |
| DG(4:0_16:1) | 6.72 | 0.029 | 1.53 | 0.15 | nc-S |
| PC(34:3) | 6.72 | 0.029 | 1.53 | 0.15 | G-nc; G-S |
| TG(4:0_16:0_16:0) | 6.61 | 0.030 | 1.52 | 0.15 | nc-S |
| PE(22:6_16:0) | 6.53 | 0.031 | 1.51 | 0.15 | G-nc; S-nc |
| DG(4:0_18:1) | 6.30 | 0.034 | 1.47 | 0.16 | nc-S |
| SPH(d18:1) | 6.28 | 0.034 | 1.47 | 0.16 | nc-G; S-G |
| PC(O-35:8) | 6.26 | 0.034 | 1.47 | 0.16 | S-nc; S-G |
| LSM(d14:1) | 6.26 | 0.034 | 1.47 | 0.16 | nc-S |
| PC(15:0_18:1) | 6.12 | 0.036 | 1.45 | 0.16 | G-nc |
| TG(4:0_14:0_16:0) | 5.93 | 0.038 | 1.42 | 0.17 | nc-S |
| PE(20:3_18:0) | 5.90 | 0.038 | 1.42 | 0.17 | G-S |
| PC(37:8) | 5.87 | 0.039 | 1.41 | 0.17 | S-G |
| TG(5:0_16:0_16:1) | 5.85 | 0.039 | 1.41 | 0.17 | nc-S |
| PC(16:0_18:3) | 5.80 | 0.040 | 1.40 | 0.17 | G-nc |
| PC(29:0) | 5.77 | 0.040 | 1.40 | 0.17 | G-nc; G-S |
| PS(18:1_18:1) | 5.70 | 0.041 | 1.39 | 0.17 | G-nc; G-S |
| PC(17:3_18:0) | 5.66 | 0.042 | 1.38 | 0.17 | G-S |
| PE(18:0_22:3) | 5.54 | 0.043 | 1.36 | 0.18 | G-S |
| PC(23:4) | 5.54 | 0.043 | 1.36 | 0.18 | S-G |
| PC(19:1_18:1) | 5.42 | 0.045 | 1.34 | 0.18 | nc-G; S-G |
| PC(32:2) | 5.39 | 0.046 | 1.34 | 0.18 | G-S |
| PC(33:2) | 5.33 | 0.047 | 1.33 | 0.18 | G-nc; G-S |

*(Continued)*

**Table 2.** (Continued)

| Lipid species | *F*-value | *P*-value | $-\log_{10}$ (*P*-value) | FDR adjusted *P*-value | Pairwise comparisons |
|---|---|---|---|---|---|
| PC(O-39:8) | 5.24 | 0.048 | 1.32 | 0.18 | G-nc; S-nc |
| SM(d38:2) | 5.21 | 0.049 | 1.31 | 0.18 | nc-G |
| PC(O-37:9) | 5.18 | 0.049 | 1.31 | 0.18 | G-nc |
| PC(14:0_16:0) | 5.17 | 0.050 | 1.31 | 0.18 | G-nc; G-S |

Data were analyzed using one-way ANOVA followed by false discovery rate (FDR) correction. Pairwise comparisons using Fisher's LSD post hoc tests were performed on species showing significant differences in ANOVA (*P* < 0.05). The comparison of lipid species quantities is presented in the format 'Higher - Lower'. For example, 'G - nc' indicates that the quantity was higher in G compared to nc.

Abbreviations: ANOVA, analysis of variance; Fisher's LSD, Fisher's least significant difference; Cer, ceramide; ChE, cholesterol ester; DG, diacylglycerol; HexCer, hexosylceramide; LSM, lysosphingomyelin; PC, phosphatidylcholine; PE, phosphatidylethanolamine; PI, phosphatidylinositol; PS, phosphatidylserine; SM, sphingomyelin; SPH, sphingosine; TG, triacylglycerol; nc, siRNA-negative control; G, siRNA-G protein subunit beta 1 (*GNB1*); S, siRNA-scavenger receptor class B member 2 (*SCARB2*); O-, alkyl acyl form; P-, alkenyl acyl form; d, dihydroxy base.

proteins that were significantly decreased following *GNB1* knockdown. Similarly, 55 proteins were significantly increased and 59 were significantly decreased following *SCARB2* knockdown (Table 4).

Data were analyzed using one-way ANOVA followed by false discovery rate (FDR) correction. The table shows 141 proteins with FDR adjusted *P* < 0.05. Pairwise comparisons using Fisher's LSD post hoc tests were performed on these proteins to determine specific group differences. The comparison of protein quantities is presented in the format 'Higher - Lower'. For example, 'G - nc' indicates that the quantity was higher in G compared to nc. Abbreviations: ANOVA: analysis of variance; Fisher's LSD: Fisher's least significant difference; nc: siRNA-negative control; G: siRNA-G protein subunit beta 1 (*GNB1*); S: siRNA-scavenger receptor class B member 2 (*SCARB2*).

The 141 proteins that showed significant changes compared to the siRNA-negative control included GNB1 and SCARB2. The heatmap of these 141 proteins is shown in Fig 6C. Specifically, in adipocytes with *GNB1* knockdown, phospholipid phosphatase 1 (PLPP1) and cadherin 13 (CDH13/T-cadherin) were significantly increased, whereas heat shock protein family A (Hsp70) member 8 (HSPA8/HSC70) was significantly decreased after FDR correction (adjusted *P* < 0.05). In contrast, in adipocytes with *SCARB2* knockdown, PLPP1 and methyltransferase-like protein 7A (METTL7A) were significantly decreased, while perilipin 2 (PLIN2), HSPA8/HSC70, NPC intracellular cholesterol transporter 2 (NPC2), and sequestosome 1 (SQSTM1/p62) were significantly increased after FDR correction (adjusted *P* < 0.05). Notably, PLPP1, involved in lipid metabolism and adipogenesis, showed contrasting results between the two knockdown conditions: it was upregulated in *GNB1* knockdown adipocytes but downregulated in *SCARB2* knockdown adipocytes. Detailed information on the significantly different proteins is presented in Table 4. Box and whisker plots of the top 141 proteins that remained significant after FDR correction (adjusted *P* < 0.05) are shown in S5 Fig. Scatterplots of *P*-values from one-way ANOVA are shown in Fig 6D, where smaller *P*-values obtained from one-way ANOVA (indicating more significant differences) correspond to larger dots (p.log) and colors transitioning from yellow to red (-$\log_{10}$ (raw *P*-value).

## Discussion

The present study evaluated the effects of *GNB1* and *SCARB2* on fat accumulation and elucidated their roles in human subcutaneous adipocytes through comprehensive

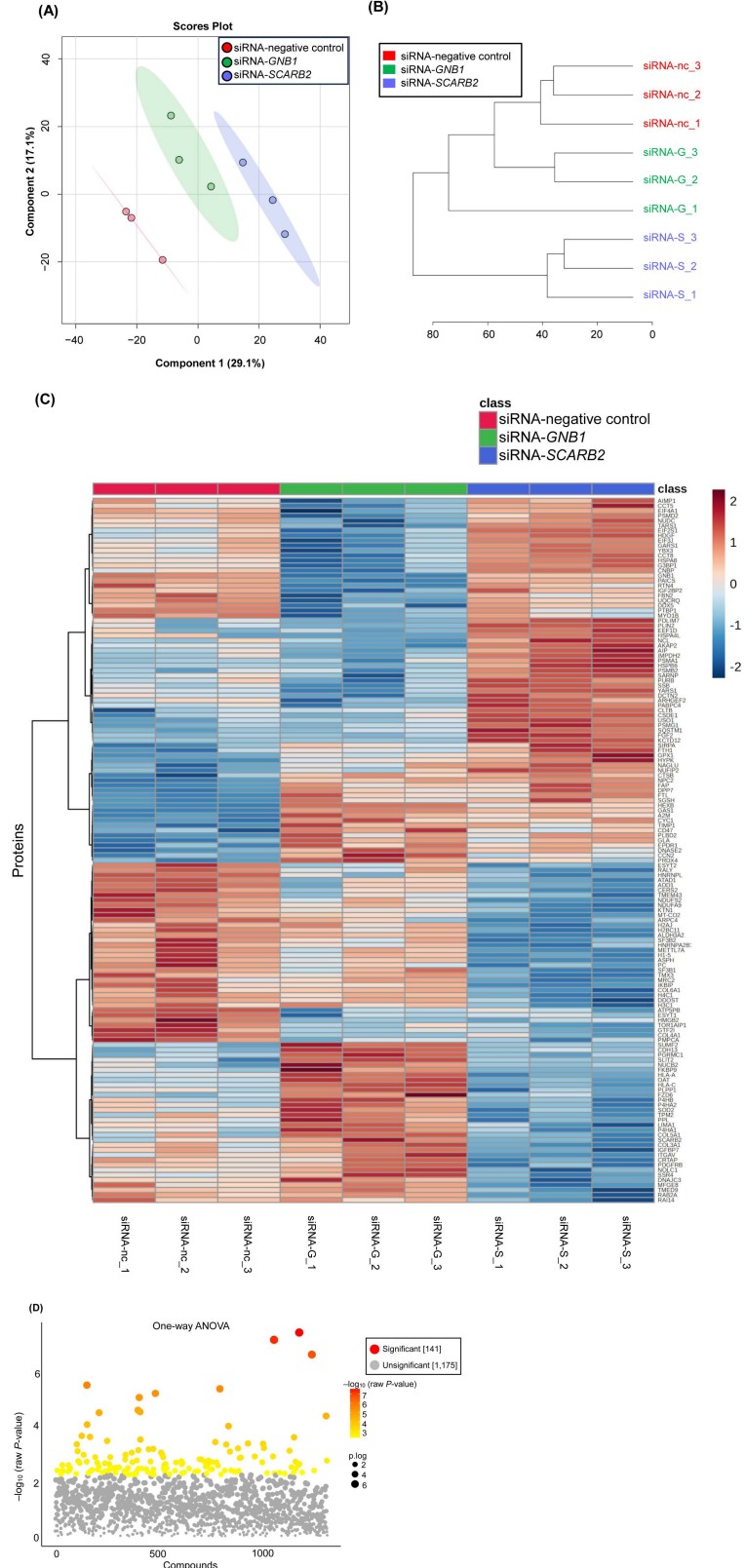

**Fig 6. Differential protein analysis in siRNA-transfected human subcutaneous adipocytes.** (A) Partial Least Squares Discriminant Analysis (PLS-DA) of proteins in siRNA-transfected human subcutaneous adipocytes. (B)

Dendrogram showing the relationship between proteins in the siRNA-transfected human subcutaneous adipocytes using Euclidean distances and Ward's clustering method. (C) Hierarchical clustering heatmap of the top 141 proteins showing differential expression after FDR correction (adjusted *P* < 0.05) in siRNA-transfected human subcutaneous adipocytes. The heatmap was generated using Ward's method with the Euclidean distance. The number at the bottom indicates the replicate number. The color scale indicates the number of standard deviations from the overall average of the proteins, with areas in dark red indicating a higher amount of the proteins, whereas areas in dark blue indicate a lower amount of the proteins. The red, green, and blue colors represent cells transfected with the siRNA-negative control, siRNA-*GNB1*, and siRNA-*SCARB2*, respectively. (D) Scatterplots of *P*-values were obtained using one-way analysis of variance (ANOVA) of 1,316 proteins in siRNA-transfected human subcutaneous adipocytes. The intensity of the red dots corresponds to the $-\log_{10}$ transformed *P*-values obtained from one-way ANOVA ($-\log_{10}$ (raw *P*-value)). FDR correction of ANOVA results identified 141 statistically significant proteins, which were further analyzed using Fisher's LSD post hoc tests to determine specific group differences. Both figures were generated using MetaboAnalyst 6.0. Abbreviations: siRNA-nc, siRNA-negative control; siRNA-G, siRNA-G protein subunit beta 1 (*GNB1*); siRNA-S, siRNA-scavenger receptor class B member 2 (*SCARB2*).

proteomic and lipidomic analyses. This study demonstrated that knockdown of *GNB1* and *SCARB2* reduces lipid accumulation in human subcutaneous adipocytes. In *GNB1* knockdown adipocytes, the expression of key adipogenic regulator genes (*PPARG* and *CEBPA*) and adipocyte-specific genes (*FABP4* and *ADIPOQ*) was maintained at normal levels. This provides evidence that *GNB1* knockdown does not impair the adipogenic program. Therefore, the observed reduction in lipid accumulation appears to be due to changes in metabolic regulation independent of differentiation. Similarly, *SCARB2* knockdown adipocytes maintained normal expression levels of these adipogenic markers, indicating that the reduced lipid accumulation in *SCARB2* knockdown is not due to impaired differentiation. These findings, together with our proteomic and lipidomic analyses, suggest that GNB1 and SCARB2 regulate adipocyte lipid accumulation through distinct pathways, consistent with their different cellular localizations and functions.

**Table 3. Summary of the number of statistically significant lipids after knockdown of *GNB1* and *SCARB2*.**

| Class | *GNB1* | | *SCARB2* | |
|---|---|---|---|---|
| | Increase | Decrease | Increase | Decrease |
| Cer | 8 (1) | 0 | 0 | 0 |
| ChE | 0 | 9 (6) | 4 (3) | 2 (1) |
| DG | 1 | 0 | 1 | 2 |
| Hex1Cer | 1 | 0 | 1 | 0 |
| LSM | 0 | 0 | 0 | 1 |
| PC | 44 (4) | 1 | 28 (3) | 1 |
| PE | 1 | 0 | 1 | 0 |
| PI | 8 (1) | 0 | 1 (1) | 0 |
| PS | 1 | 0 | 0 | 0 |
| SM | 0 | 1 | 0 | 0 |
| SPH | 0 | 1 | 0 | 0 |
| TG | 0 | 0 | 0 | 6 |

The number of lipids with *P* < 0.05 from one-way ANOVA is shown, with the number of lipids that remained significant after FDR correction (adjusted *P* < 0.05) indicated in parentheses. Abbreviations: ANOVA, analysis of variance; Fisher's LSD, Fisher's least significant difference; Cer, ceramide; ChE, cholesterol ester; DG, diacylglycerol; HexCer, hexosylceramide; LSM, lysosphingomyelin; PC, phosphatidylcholine; PE, phosphatidylethanolamine; PI, phosphatidylinositol; PS, phosphatidylserine; SM, sphingomyelin; SPH, sphingosine; TG, triacylglycerol.

**Table 4. Summary of statistically significant proteins after knockdown of *GNB1* and *SCARB2*.**

| Gene name | F-value | P-value | $-\log_{10}$ (P-value) | FDR adjusted P-value | Pairwise comparisons |
|---|---|---|---|---|---|
| ALDH3A2 | 978.56 | $2.86 \times 10^{-8}$ | 7.54 | $3.51 \times 10^{-5}$ | nc-G; nc-S; G-S |
| GAS1 | 794.11 | $5.33 \times 10^{-8}$ | 7.27 | $3.51 \times 10^{-5}$ | G-nc; S-nc; G-S |
| KCTD12 | 521.03 | $1.88 \times 10^{-7}$ | 6.73 | $8.23 \times 10^{-5}$ | S-nc; S-G |
| HEXB | 216.79 | $2.54 \times 10^{-6}$ | 5.59 | $8.37 \times 10^{-4}$ | G-nc; S-nc; G-S |
| PAICS | 194.92 | $3.48 \times 10^{-6}$ | 5.46 | $9.17 \times 10^{-4}$ | nc-G; S-G |
| GNB1 | 170.95 | $5.13 \times 10^{-6}$ | 5.29 | 0.0011 | nc-G; nc-S; S-G |
| HLA-C | 151.58 | $7.31 \times 10^{-6}$ | 5.14 | 0.0014 | G-nc; nc-S; G-S |
| OAT | 105.00 | $2.14 \times 10^{-5}$ | 4.67 | 0.0035 | G-nc; nc-S; G-S |
| HLA-A | 99.83 | $2.48 \times 10^{-5}$ | 4.60 | 0.0035 | G-nc; nc-S; G-S |
| NPC2 | 97.47 | $2.66 \times 10^{-5}$ | 4.57 | 0.0035 | G-nc; S-nc |
| FGF2 | 88.40 | $3.54 \times 10^{-5}$ | 4.45 | 0.0042 | S-nc; S-G |
| AKAP2 | 68.59 | $7.36 \times 10^{-5}$ | 4.13 | 0.0081 | nc-G; S-nc; S-G |
| CDH13 | 65.58 | $8.37 \times 10^{-5}$ | 4.08 | 0.0085 | G-nc; G-S |
| PGRMC1 | 48.91 | $1.93 \times 10^{-4}$ | 3.71 | 0.018 | G-nc; G-S |
| SUMF2 | 47.46 | $2.10 \times 10^{-4}$ | 3.68 | 0.018 | G-nc; G-S |
| PSMG1 | 46.85 | $2.18 \times 10^{-4}$ | 3.66 | 0.018 | S-nc; S-G |
| PLIN2 | 43.76 | $2.64 \times 10^{-4}$ | 3.58 | 0.020 | S-nc; S-G |
| SQSTM1 | 41.95 | $2.97 \times 10^{-4}$ | 3.53 | 0.022 | S-nc; S-G |
| TARS1 | 40.59 | $3.26 \times 10^{-4}$ | 3.49 | 0.023 | nc-G; S-nc; S-G |
| USO1 | 38.86 | $3.68 \times 10^{-4}$ | 3.43 | 0.023 | S-nc; S-G |
| UQCRQ | 38.56 | $3.76 \times 10^{-4}$ | 3.42 | 0.023 | nc-G; S-G |
| H4C1 | 37.89 | $3.95 \times 10^{-4}$ | 3.40 | 0.023 | nc-S; G-S |
| YARS1 | 37.67 | $4.01 \times 10^{-4}$ | 3.40 | 0.023 | nc-G; S-nc; S-G |
| NDUFA9 | 33.85 | $5.40 \times 10^{-4}$ | 3.27 | 0.029 | nc-G; nc-S; G-S |
| A2M | 33.38 | $5.61 \times 10^{-4}$ | 3.25 | 0.029 | G-nc; S-nc |
| HSPB6 | 32.91 | $5.83 \times 10^{-4}$ | 3.23 | 0.029 | S-nc; S-G |
| IMPDH2 | 32.48 | $6.04 \times 10^{-4}$ | 3.22 | 0.029 | S-nc; S-G |
| PABPC4 | 31.58 | $6.53 \times 10^{-4}$ | 3.18 | 0.031 | nc-G; S-nc; S-G |
| HYPK | 31.06 | $6.84 \times 10^{-4}$ | 3.17 | 0.031 | G-nc; S-nc; S-G |
| H2BC11 | 30.68 | $7.07 \times 10^{-4}$ | 3.15 | 0.031 | nc-S; G-S |
| NDUFS2 | 30.21 | $7.37 \times 10^{-4}$ | 3.13 | 0.031 | nc-G; nc-S; G-S |
| RAB2A | 29.27 | $8.03 \times 10^{-4}$ | 3.10 | 0.032 | nc-S; G-S |
| G3BP1 | 29.25 | $8.05 \times 10^{-4}$ | 3.09 | 0.032 | nc-G; S-nc; S-G |
| DCTN2 | 27.54 | $9.48 \times 10^{-4}$ | 3.02 | 0.035 | nc-G; S-nc; S-G |
| PSMD2 | 27.38 | $9.63 \times 10^{-4}$ | 3.02 | 0.035 | nc-G; S-G |
| SSB | 27.34 | $9.66 \times 10^{-4}$ | 3.01 | 0.035 | S-nc; S-G |
| MT-CO2 | 25.97 | 0.0011 | 2.95 | 0.038 | nc-G; nc-S; G-S |
| IKBIP | 25.96 | 0.0011 | 2.95 | 0.038 | nc-S; G-S |
| HDGF | 25.42 | 0.0012 | 2.93 | 0.038 | nc-G; S-nc; S-G |
| ASPH | 25.38 | 0.0012 | 2.93 | 0.038 | nc-S; G-S |
| CNBP | 25.36 | 0.0012 | 2.93 | 0.038 | nc-G; S-nc; S-G |
| COL4A1 | 25.06 | 0.0012 | 2.91 | 0.038 | nc-G; nc-S |
| NUFIP2 | 24.73 | 0.0013 | 2.90 | 0.039 | G-nc; S-nc; S-G |
| PDGFRB | 23.76 | 0.0014 | 2.85 | 0.040 | nc-S; G-S |
| SSR4 | 23.59 | 0.0014 | 2.84 | 0.040 | G-nc; nc-S; G-S |

*(Continued)*

**Table 4.** (Continued)

| Gene name | F-value | P-value | −log₁₀ (P-value) | FDR adjusted P-value | Pairwise comparisons |
|---|---|---|---|---|---|
| *RTN4* | 23.56 | 0.0014 | 2.84 | 0.040 | nc-G; S-G |
| *IGF2 BP2* | 22.86 | 0.0016 | 2.81 | 0.040 | nc-G; S-G |
| *MFGE8* | 22.78 | 0.0016 | 2.80 | 0.040 | nc-S; G-S |
| *PLPP1* | 22.67 | 0.0016 | 2.80 | 0.040 | G-nc; nc-S; G-S |
| *H1-5* | 22.37 | 0.0017 | 2.78 | 0.040 | nc-S; G-S |
| *IGFBP7* | 22.23 | 0.0017 | 2.77 | 0.040 | nc-S; G-S |
| *NOLC1* | 22.07 | 0.0017 | 2.77 | 0.040 | G-nc; nc-S; G-S |
| *AIP* | 22.05 | 0.0017 | 2.77 | 0.040 | S-nc; S-G |
| *FBN2* | 22.04 | 0.0017 | 2.76 | 0.040 | nc-G; S-G |
| *PRDX4* | 21.93 | 0.0017 | 2.76 | 0.040 | G-nc; S-nc; G-S |
| *PDLIM7* | 21.66 | 0.0018 | 2.74 | 0.040 | S-nc; S-G |
| *GARS1* | 21.62 | 0.0018 | 2.74 | 0.040 | nc-G; S-G |
| *TIMP1* | 21.54 | 0.0018 | 2.74 | 0.040 | G-nc; S-nc |
| *TMED9* | 21.50 | 0.0018 | 2.74 | 0.040 | nc-S; G-S |
| *RALY* | 21.43 | 0.0019 | 2.73 | 0.040 | nc-G; nc-S; G-S |
| *H2AJ* | 21.26 | 0.0019 | 2.72 | 0.040 | nc-S; G-S |
| *DDX5* | 21.23 | 0.0019 | 2.72 | 0.040 | nc-G; S-G |
| *P4HA1* | 21.03 | 0.0019 | 2.71 | 0.041 | G-nc; G-S |
| *P4HA2* | 20.86 | 0.0020 | 2.70 | 0.041 | G-nc; nc-S; G-S |
| *METTL7A* | 20.40 | 0.0021 | 2.68 | 0.042 | nc-S; G-S |
| *CCN2* | 20.39 | 0.0021 | 2.68 | 0.042 | G-nc; S-nc; G-S |
| *DPP7* | 20.34 | 0.0021 | 2.67 | 0.042 | G-nc; S-nc |
| *CYC1* | 20.24 | 0.0022 | 2.67 | 0.042 | G-nc; S-nc |
| *HMGB2* | 20.13 | 0.0022 | 2.66 | 0.042 | nc-G; nc-S |
| *HSPA8* | 19.71 | 0.0023 | 2.64 | 0.042 | nc-G; S-nc; S-G |
| *ESYT2* | 19.71 | 0.0023 | 2.64 | 0.042 | nc-G; nc-S; G-S |
| *COL3A1* | 19.67 | 0.0023 | 2.63 | 0.042 | nc-S; G-S |
| *SLIT2* | 19.63 | 0.0023 | 2.63 | 0.042 | G-nc; G-S |
| *DNASE2* | 19.56 | 0.0024 | 2.63 | 0.042 | G-nc; S-nc |
| *DNAJC3* | 19.34 | 0.0024 | 2.62 | 0.043 | nc-S; G-S |
| *ITGAV* | 18.76 | 0.0026 | 2.58 | 0.045 | nc-S; G-S |
| *GLA* | 18.71 | 0.0026 | 2.58 | 0.045 | G-nc; S-nc |
| *GTF2I* | 18.38 | 0.0028 | 2.56 | 0.045 | nc-G; nc-S |
| *PC* | 18.20 | 0.0028 | 2.55 | 0.045 | nc-G; nc-S; G-S |
| *HNRNPL* | 18.14 | 0.0029 | 2.54 | 0.045 | nc-G; nc-S |
| *FAP* | 17.91 | 0.0030 | 2.53 | 0.045 | G-nc; S-nc |
| *SOD2* | 17.86 | 0.0030 | 2.53 | 0.045 | G-nc; nc-S; G-S |
| *EIF2S1* | 17.81 | 0.0030 | 2.52 | 0.045 | S-nc; S-G |
| *TOR1AIP1* | 17.72 | 0.0030 | 2.52 | 0.045 | nc-G; nc-S |
| *CERS2* | 17.59 | 0.0031 | 2.51 | 0.045 | nc-G; nc-S; G-S |
| *SCARB2* | 17.58 | 0.0031 | 2.51 | 0.045 | G-nc; nc-S; G-S |
| *ADD1* | 17.51 | 0.0031 | 2.50 | 0.045 | nc-G; nc-S; G-S |
| *TMX3* | 17.48 | 0.0031 | 2.50 | 0.045 | nc-S; G-S |
| *ATAD1* | 17.41 | 0.0032 | 2.50 | 0.045 | nc-G; nc-S; G-S |
| *CCT8* | 17.34 | 0.0032 | 2.49 | 0.045 | nc-G; S-G |

*(Continued)*

**Table 4.** (Continued)

| Gene name | *F*-value | *P*-value | $-\log_{10}$ (*P*-value) | FDR adjusted *P*-value | Pairwise comparisons |
|---|---|---|---|---|---|
| NAGLU | 17.34 | 0.0032 | 2.49 | 0.045 | G-nc; S-nc |
| CCT5 | 17.14 | 0.0033 | 2.48 | 0.045 | nc-G; S-G |
| EPDR1 | 17.07 | 0.0033 | 2.48 | 0.045 | G-nc; S-nc |
| PURB | 17.06 | 0.0033 | 2.48 | 0.045 | S-nc; S-G |
| KTN1 | 16.97 | 0.0034 | 2.47 | 0.045 | nc-S; G-S |
| CTSB | 16.85 | 0.0035 | 2.46 | 0.045 | G-nc; S-nc |
| FTH1 | 16.77 | 0.0035 | 2.46 | 0.045 | S-nc; S-G |
| NUDC | 16.56 | 0.0036 | 2.44 | 0.045 | nc-G; S-G |
| CSDE1 | 16.43 | 0.0037 | 2.43 | 0.045 | S-nc; S-G |
| P4HB | 16.43 | 0.0037 | 2.43 | 0.045 | G-nc; G-S |
| PPL | 16.43 | 0.0037 | 2.43 | 0.045 | G-nc; nc-S; G-S |
| SF3B1 | 16.41 | 0.0037 | 2.43 | 0.045 | nc-S; G-S |
| TPM2 | 16.29 | 0.0038 | 2.42 | 0.045 | G-nc; nc-S; G-S |
| EIF4A1 | 16.29 | 0.0038 | 2.42 | 0.045 | nc-G; S-G |
| EEF1D | 16.27 | 0.0038 | 2.42 | 0.045 | S-nc; S-G |
| MRC2 | 16.22 | 0.0038 | 2.42 | 0.045 | nc-S; G-S |
| COL6A1 | 16.08 | 0.0039 | 2.41 | 0.045 | nc-S; G-S |
| EIF3J | 16.07 | 0.0039 | 2.41 | 0.045 | nc-G; S-G |
| DDOST | 16.02 | 0.0039 | 2.41 | 0.045 | nc-S; G-S |
| ARHGEF2 | 16.01 | 0.0039 | 2.41 | 0.045 | nc-G; S-nc; S-G |
| ESYT1 | 16.01 | 0.0039 | 2.41 | 0.045 | nc-G; nc-S |
| SGSH | 15.99 | 0.0039 | 2.40 | 0.045 | G-nc; S-nc |
| LIMA1 | 15.96 | 0.0040 | 2.40 | 0.045 | G-nc; nc-S; G-S |
| GPX1 | 15.94 | 0.0040 | 2.40 | 0.045 | G-nc; S-nc; S-G |
| HNRNPA2B1 | 15.89 | 0.0040 | 2.40 | 0.045 | nc-S; G-S |
| ARPC4 | 15.88 | 0.0040 | 2.40 | 0.045 | nc-G; nc-S; G-S |
| CRTAP | 15.85 | 0.0040 | 2.39 | 0.045 | nc-S; G-S |
| SARNP | 15.76 | 0.0041 | 2.39 | 0.046 | S-nc; S-G |
| ATP5PB | 15.69 | 0.0041 | 2.38 | 0.046 | nc-G; nc-S |
| CLTB | 15.40 | 0.0043 | 2.36 | 0.047 | S-nc; S-G |
| FTL | 15.40 | 0.0043 | 2.36 | 0.047 | G-nc; S-nc |
| PSMB2 | 15.37 | 0.0044 | 2.36 | 0.047 | S-nc; S-G |
| FZD6 | 15.31 | 0.0044 | 2.36 | 0.047 | G-nc; G-S |
| CD47 | 15.25 | 0.0044 | 2.35 | 0.047 | G-nc; S-nc; G-S |
| PSMA1 | 15.20 | 0.0045 | 2.35 | 0.047 | S-nc; S-G |
| FKBP9 | 15.18 | 0.0045 | 2.35 | 0.047 | G-nc; G-S |
| HSPA4L | 15.08 | 0.0046 | 2.34 | 0.047 | S-nc; S-G |
| SIRPA | 14.99 | 0.0046 | 2.33 | 0.047 | G-nc; S-nc |
| PLBD2 | 14.97 | 0.0047 | 2.33 | 0.047 | G-nc; S-nc |
| COL5A1 | 14.93 | 0.0047 | 2.33 | 0.047 | G-nc; G-S |
| PMPCA | 14.83 | 0.0048 | 2.32 | 0.048 | nc-G; nc-S |
| RAI14 | 14.82 | 0.0048 | 2.32 | 0.048 | nc-S; G-S |
| YBX3 | 14.66 | 0.0049 | 2.31 | 0.048 | nc-G; S-G |
| SF3B2 | 14.63 | 0.0049 | 2.31 | 0.048 | nc-S; G-S |
| MYO1B | 14.60 | 0.0050 | 2.31 | 0.048 | nc-G; S-G |

*(Continued)*

**Table 4.** (Continued)

| Gene name | F-value | P-value | −log₁₀ (P-value) | FDR adjusted P-value | Pairwise comparisons |
|-----------|---------|---------|------------------|----------------------|----------------------|
| *NCL* | 14.55 | 0.0050 | 2.30 | 0.048 | S-nc; S-G |
| *AIMP1* | 14.53 | 0.0050 | 2.30 | 0.048 | nc-G; S-G |
| *NUCB2* | 14.38 | 0.0051 | 2.29 | 0.049 | G-nc; G-S |
| *TMEM43* | 14.35 | 0.0052 | 2.29 | 0.049 | nc-G; nc-S |
| *H3C1* | 14.34 | 0.0052 | 2.29 | 0.049 | nc-S; G-S |
| *PTBP1* | 14.25 | 0.0053 | 2.28 | 0.049 | nc-G; S-G |

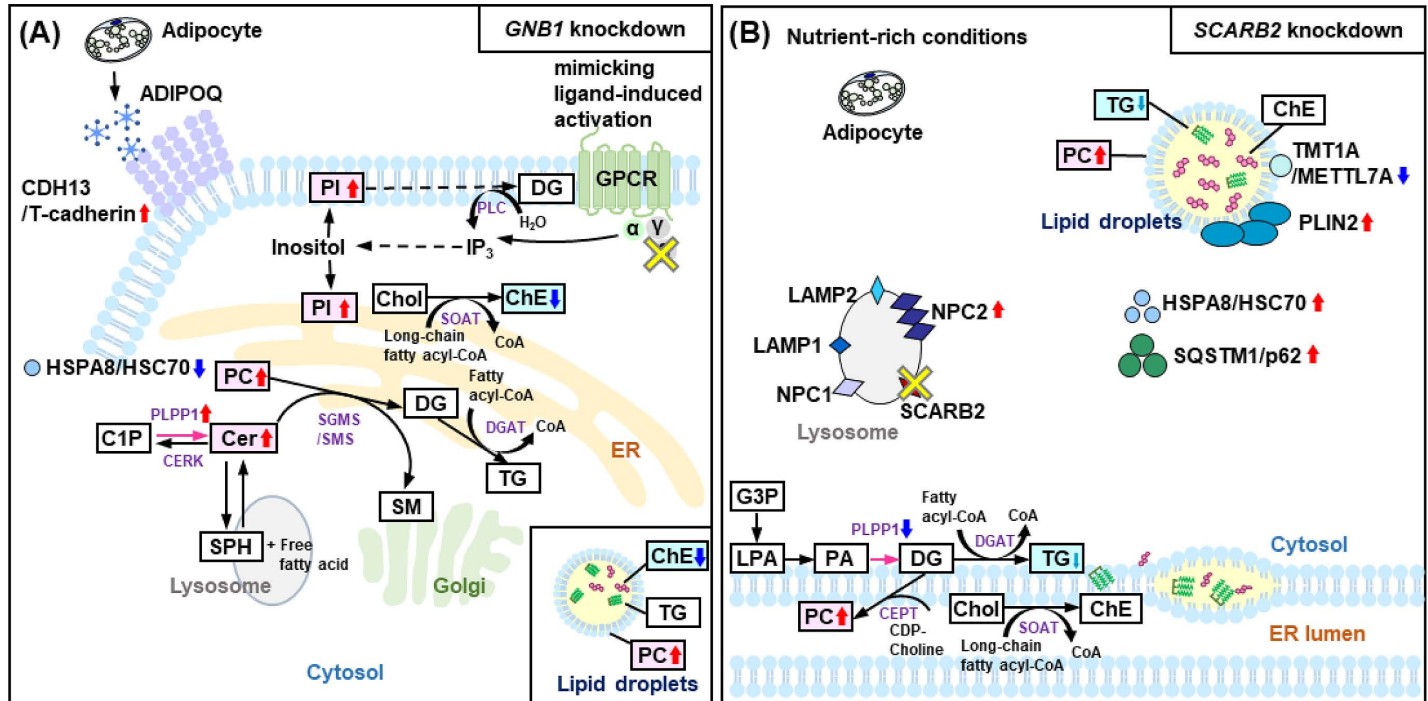

**Fig 7. Schematic model depicting the relationship between lipids and proteins by target gene knockdown.** (A) Changes observed following the knockdown of G protein subunit beta 1 (*GNB1*). (B) Effects observed after the knockdown of *SCARB2*. Although both *GNB1* and *SCARB2* knockdown results in fewer lipid droplets in the cytoplasm under nutrient-rich conditions, the model emphasizes significant changes in lipid and protein states. In the schematic diagram, significant increases are indicated by red arrows pointing upwards, while significant decreases are shown with blue arrows pointing downwards. Non-significant but decreasing trends are marked with small light blue arrows pointing downwards. Abbreviations: ADIPOQ, adiponectin, C1Q and collagen domain containing; α, alpha subunit of the G proteins; β, GNB1; C1P, ceramide-1-phosphate; Cer, ceramide; CERK, ceramide kinase; CDH13/T-cadherin, cadherin 13; CEPT, choline/ethanolaminephosphotransferase; ChE, cholesterol ester; Chol, cholesterol; DG, diacylglycerol; DGAT, diacylglycerol O-acyltransferase; ER, endoplasmic reticulum; G3P, glycero-3-phosphate; GPCR, G protein-coupled receptor; γ, gamma subunit of the G proteins; HSPA8/HSC70, heat shock protein family A; IP₃, inositol (1,4,5)-trisphosphate; LAMP1/2, lysosomal associated membrane protein 1/2; LPA, lysophosphatidic acid; NPC1/2, NPC intracellular cholesterol transporter 1/2; PA, phosphatidic acid; PC, phosphatidylcholine; PI, phosphatidylinositol; PLIN2, perilipin 2; PLPP1, phospholipid phosphatase 1; PLC, phospholipase C; PNPLA2/ATGL, patatin like phospholipase domain containing 2; SCARB2, scavenger receptor class B member 2; SGMS/SMS, sphingomyelin synthase; SM, sphingomyelin; SOAT, sterol O-acyltransferase; SPH, sphingosine; SQSTM1/p62, sequestosome 1; TG, triacylglycerol; TMT1A/METTL7A, methyltransferase-like protein 7A.

## Effects of *GNB1* knockdown

The knockdown of *GNB1* significantly altered lipid and protein profiles in human subcutaneous adipocytes (Fig 7A).

Our lipidomic analysis indicated that the knockdown of *GNB1* led to a significant increase in PC and PI levels. Both PC and PI are crucial for membrane structure and cellular signaling [46]. The increase in PC levels could reflect changes in membrane composition, which may impact adipocyte function and fat accumulation. PI in the cell membrane is synthesized into PI 4,5-bisphosphate through a two-step reaction catalyzed by lipid kinases [47]. Subsequently, the enzyme phospholipase hydrolyzes the PI 4,5-bisphosphate in the cell membrane, converting it to $IP_3$ and DG [48]. In light of these findings, the elevated PI levels suggest an alteration in the $IP_3$/DG signaling pathway. This change may mimic ligand-induced activation, potentially enhancing signaling through this pathway and affecting lipid composition and adipocyte function. In contrast, ChE levels were significantly reduced in *GNB1* knockdown adipocytes. ChEs, which participate in lipid storage and transport rather than membrane structure, are typically sequestered in lipid droplets [49]. The observed decrease in ChE levels may influence lipid droplet formation and fat accumulation in these cells.

Proteomic analysis also showed notable changes in several key proteins involved in lipid metabolism and adipocyte function. In particular, PLPP1 and CDH13/T-cadherin expression levels were significantly increased, while that of HSPA8/HSC70 was significantly decreased following *GNB1* knockdown in adipocytes. PLPP1 plays a role in the conversion of ceramide-1-phosphate to Cer [50]. The observed increase in Cer following the knockdown of *GNB1* suggests that *GNB1* may regulate Cer metabolism by modulating PLPP1 expression.

CDH13/T-cadherin, a member of the cadherin superfamily, is associated with cell-cell adhesion and adiponectin-mediated metabolic processes [51]. Additionally, ADIPOQ, a key circulating protein known as a physiologically active substance is produced exclusively by adipocytes. Although statistical analysis did not reveal a significant difference in ADIPOQ levels, the observed increase in CDH13/T-cadherin suggests potential implications for adiponectin-mediated metabolic processes.

HSPA8/HSC70, a member of the heat shock protein family, serves as an exosome marker [52]. A recent study has highlighted the role of the adiponectin/T-cadherin system in enhancing exosome biogenesis and secretion. Their findings indicate that adiponectin aids in the removal of excess Cer by facilitating its transport to exosomes, thereby reducing Cer accumulation in endothelial cells. Knockdown of *GNB1* led to a significant downregulation of HSPA8/HSC70. A reduction in the expression of HSPA8/HSC70 could impair the effective cellular secretion of Cer-containing exosomes. This disruption in Cer removal pathways may result in an intracellular accumulation of Cer, which aligns with the observed increase in Cer levels (Fig 7A).

## Effects of *SCARB2* knockdown

Proteomic analysis revealed significant alterations in several key proteins associated with lipid synthesis, lipid droplet localization, and adipocyte function. Specifically, PLPP1 and METTL7A were significantly downregulated, whereas PLIN2, HSPA8/HSC70, NPC2, and SQSTM1/p62 were significantly upregulated following *SCARB2* knockdown in adipocytes.

Assessment of lipid droplet formation and the enzymes involved in neutral lipid synthesis, including TG, revealed that *SCARB2* knockdown with siRNA led to the downregulation of PLPP1, which is essential for the conversion of PA to DG. PLPP1 enzymes are typically localized on the plasma membrane, endoplasmic reticulum (ER), and Golgi apparatus. In the ER, they contribute to DG production for glycerolipid synthesis [50,53]. The reduction in PLPP1

may consequently decrease DG production, which could affect TG levels within the ER and contribute to the observed decrease in lipid droplet formation.

Thiol methyltransferase 1A (TMT1A; also known as METTL7A and AAM-B) [54] is localized to lipid droplets in the droplet-containing cells and is lost from the lipid droplets with their regression [55]. In *SCARB2* knockdown adipocytes, TMT1A/METTL7A expression was decreased, correlating with a reduced number of lipid droplets in the cytoplasm. This suggests a potential reduction in the level of integral droplet proteins such as TMT1A.

Perilipins, specifically PLIN2, are proteins with a PAT domain involved in the regulation of lipid droplet accumulation and degradation. PLIN2 plays a significant role in intracellular lipid metabolism and is associated with obesity. *SCARB2* knockdown resulted in a significant increase in PLIN2 levels. Given that perilipin content in adipocytes from obese individuals is typically low [56], the increase in PLIN2 observed in *SCARB2* knockdown cells suggests that *SCARB2* knockdown could potentially mitigate obesity risk by enhancing lipid droplet regulation and reducing fat accumulation.

Autophagy is a cellular process that maintains homeostasis by degrading proteins, lipids, and organelles in lysosomes, which then recycle the degraded products to provide energy [57, 58]. PLIN2 contains a recognition sequence for HSPA8/HSC70 and is targeted for degradation via chaperone-mediated autophagy (CMA) [59]. Additionally, sequestosome 1 (SQSTM1), also known as p62, plays a key role in autophagy [60]. In *SCARB2* knockdown cells, both HSPA8/HSC70 and SQSTM1 levels were significantly increased. As illustrated in Fig 7, CMA is activated under conditions of prolonged starvation [61], mild oxidative stress [62], and hypoxia [63]. We investigated phosphorylation modifications using Proteome Discoverer; however, phosphorylation of PLIN2 and SQSTM1 could not be confirmed. This might be due to the nutrient-rich conditions of our cell culture, which may have precluded phosphorylation modifications alongside lipolysis, CMA, and macrolipophagy. Recent studies have shown that glucose deprivation leads to the binding of choline kinase α2 (CHKα2) to PLIN2 and the subsequent CHKα2-mediated phosphorylation of PLIN2 at tyrosine 232 (Y232) [64]. Moreover, NPC2, a protein primarily involved in the export of cholesterol from lysosomes [65], was upregulated in *SCARB2* knockdown cells. This increase in NPC2 expression might indicate alterations in cholesterol homeostasis within human subcutaneous adipocytes (Fig 7B).

In summary, the knockdown of *GNB1* and *SCARB2* influenced lipid and protein metabolism through distinct pathways, highlighting their crucial roles in regulating human subcutaneous adipocyte function and fat accumulation. Our findings are supported by the previous analysis of Mardinoglu et al. [66], who analyzed gene expression profiles from the Swedish Obese Subjects Sib Pair study (GSE27916). Their study included subcutaneous adipose tissue samples from 209 female and 95 male subjects, categorized into lean (BMI < 25), overweight (25 ≤ BMI < 30), and obese (30 ≤ BMI) groups. The analysis revealed significant upregulation of both *GNB1* and *SCARB2* in the subcutaneous adipose tissue of overweight and obese subjects compared to that of lean controls (*GNB1*: adjusted $P = 0.045$ in males and $3.44 \times 10^{-6}$ in females; *SCARB2*: adjusted $P = 0.049$ in males and $4.05 \times 10^{-12}$ in females). While the stronger statistical significance observed in female subjects might reflect sex-specific regulation of adipose tissue metabolism, the difference in the sample sizes between the sexes should also be considered. These findings from human adipose tissue samples complement our in vitro results, further supporting the involvement of *GNB1* and *SCARB2* in subcutaneous adipose tissue metabolism. Future studies should explore the relationship between type 2 diabetes mellitus and the expression of these genes in subcutaneous adipose tissue, as metabolic conditions often coexist with obesity.

These findings from human subjects underscore the importance of integrating multiple data sources to understand the complex biology of obesity. GWAS data on BMI integrated with our proteomic and lipidomic findings offer a more comprehensive understanding of the genetic influences on fat accumulation. The identification of *GNB1* and *SCARB2* as potential regulators in human subcutaneous adipocytes underscores their importance in obesity-related processes, such as fat accumulation. Our findings support the hypothesis that genes associated with BMI play significant roles in fat accumulation. This integration of "omics" data provides a comprehensive view of the complex regulatory networks involved in fat accumulation, potentially revealing new targets for obesity therapy.

While our study provides valuable insights, several limitations should be acknowledged. Primarily, our use of subcutaneous adipocytes from a single donor limits the generalization of our findings. Future studies should include multiple donors to account for individual variation. Additionally, while our *in vitro* model using human subcutaneous adipocytes is highly relevant and valuable, it may not fully recapitulate the complexities of adipose tissue *in vivo*. The absence of additional functional experiments limits our ability to directly assess the physiological effects of *GNB1* and *SCARB2* knockdown. Moreover, although our end-point analysis at Day 14 provided valuable insights into the cumulative effects of *GNB1* and *SCARB2* knockdown, analyses at earlier time points could have provided additional information about the temporal dynamics of these effects. Future studies should address this aspect by performing time-course analyses of gene and protein expression across multiple stages of differentiation and fat accumulation. Lastly, while we used specific siRNAs, the possibility of off-target effects cannot be completely excluded.

Despite these limitations, our study provides important preliminary insights into the roles of *GNB1* and *SCARB2* in adipocyte function and fat accumulation. Addressing these limitations in future research will contribute to a more comprehensive understanding of these genes in metabolic processes.

## Conclusions

This study underscores the importance of *GNB1* and *SCARB2* in regulating lipid metabolism and fat accumulation in human subcutaneous adipocytes. The observed alterations in lipid and protein profiles following gene knockdown offer valuable insights into the molecular mechanisms underlying obesity, especially concerning fat accumulation. Future research should focus on elucidating the specific pathways and functions affected by these genes to identify novel therapeutic strategies for obesity and related metabolic disorders.

## Supporting information

**S1 Fig. Expression profiles of GNB1 and SCARB2 during subcutaneous adipocyte differentiation and maturation.**
(DOCX)

**S2 Fig. Visualization of target gene amplification with absolute quantification analysis.**
(DOCX)

**S3 Fig. Comparison of adipogenic marker expression in *GNB1* and *SCARB2* knockdown cells using quantitative real-time PCR.**
(DOCX)

**S4 Fig. Analysis of significant lipid species alterations following knockdown of *GNB1* and *SCARB2*.**
(PDF)

**S5 Fig. Analysis of significant protein alterations following knockdown of *GNB1* and *SCARB2*.**
(PDF)

**S1 Table. Primer sequences used for digital PCR and quantitative real-time PCR, and the estimated amplicon sizes.**
(DOCX)

**S2 Table. Digital PCR analysis program.**
(DOCX)

**S3 Table. dMIQE checklist for digital PCR experiments.**
(DOCX)

**S4 Table. Lipidomics data.** Lipid species identified by LipidSearch 5.1.9.
(XLSX)

**S5 Table. CSV file for lipidomic analysis by MetaboAnalyst 6.0.** Lipid species and normalized area values.
(CSV)

**S6 Table. Proteomics data.** Proteins identified by Proteome Discoverer 2.2.
(XLSX)

**S7 Table. CSV file for proteomic analysis by MetaboAnalyst 6.0.** Proteins and normalized abundance values.
(CSV)

**S8 Table. Data analysis of digital PCR after gene knockdown.**
(DOCX)

**S9 Table. Data analysis of quantitative real-time PCR for adipogenic marker genes in *GNB1* and *SCARB2* knockdown cells.**
(DOCX)

## Acknowledgments

We thank Dr. Minako Kondo, Dr. Makoto Goda, Dr. Asuka Miyagi, and Assoc. Prof. Isao Kosugi for their technical support and Yoshiko Ishizuka-Katsura for her valuable advice. The images of the dish and reagent bottle in Fig 2 were provided by the DBCLS Togo Picture Gallery (c2016 DBCLS TogoTV/CC-BY-4.0; https://togotv.dbcls.jp/pics.html). We would like to thank Editage (www.editage.com) for English language editing.

## Author contributions

**Conceptualization:** Aya Kitamoto, Takuya Kitamoto.

**Formal analysis:** Aya Kitamoto, Takuya Kitamoto.

**Investigation:** Aya Kitamoto, Takuya Kitamoto.

**Methodology:** Takuya Kitamoto.

**Project administration:** Aya Kitamoto.

**Software:** Takuya Kitamoto.

**Validation:** Aya Kitamoto.

**Writing – original draft:** Aya Kitamoto, Takuya Kitamoto.

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
