## [Decision Letter · Decision Letter 0]

14 Nov 2024

PONE-D-24-41568Integrative proteomic and lipidomic analysis of GNB1 and SCARB2 knockdown in human subcutaneous adipocytesPLOS ONE

Dear Dr. Kitamoto,

Thank you for submitting your manuscript to PLOS ONE. After careful consideration, we feel that it has merit but does not fully meet PLOS ONE’s publication criteria as it currently stands. Therefore, we invite you to submit a revised version of the manuscript that addresses the points raised during the review process.

We look forward to receiving your revised manuscript.

Kind regards,

Juan J Loor

Academic Editor

PLOS ONE

Journal Requirements:

2. Thank you for stating the following financial disclosure: [This work was supported by the Japan Society for the Promotion of Science KAKENHI (grant number: JP23K10957).]. Please state what role the funders took in the study. If the funders had no role, please state: "The funders had no role in study design, data collection and analysis, decision to publish, or preparation of the manuscript." If this statement is not correct you must amend it as needed. Please include this amended Role of Funder statement in your cover letter; we will change the online submission form on your behalf.

Reviewers' comments:

Reviewer's Responses to Questions

**Comments to the Author**

1. Is the manuscript technically sound, and do the data support the conclusions?

Reviewer #1: Yes

2. Has the statistical analysis been performed appropriately and rigorously? 

Reviewer #1: Yes

3. Have the authors made all data underlying the findings in their manuscript fully available?

Reviewer #1: Yes

4. Is the manuscript presented in an intelligible fashion and written in standard English?

Reviewer #1: Yes

5. Review Comments to the Author

Reviewer #1: Aya Kitamoto and Takuya Kitamoto's manuscript aims to investigate the role of GNB1 (G protein subunit beta 1) and SCARB2 (lysosomal integral membrane protein 2) in adipocyte function through integrative proteomic and lipidomic analysis. Their findings suggest that GNB1 and SCARB2 play crucial roles in regulating adipocyte functioning and lipid metabolism. While the data provided support the importance of these proteins in adipocyte function, further studies are needed to fully understand the mechanisms by which GNB1 and SCARB2 influence lipid metabolism. The data is clean, the conclusions drawn are significant, and the paper is well written and informative. However, the following concerns need to be addressed:

1. The selection of the two proteins is based on a previous study that showed that these two proteins exhibited remarkable and continuous expression changes during adipogenic differentiation. The authors do not mention whether the expression of these proteins was up-or-down regulated during the differentiation process. The authors should evaluate the expression of these proteins during the differentiation of primary adipocytes, preferably at different time points.

2. Is the expression of GNB1 and SCARB2 altered in the adipose tissue of human subjects with obesity and type 2 diabetes (T2DM)? The authors may analyze publicly available RNA sequencing datasets to show the impact of obesity and T2DM on the expression of these proteins in adipose tissue depots.

3. The knockdown of GNB1 and SCARB2 is done using multiple transfections by specific siRNAs, and the expression of these mRNAs is evaluated at the end (Day 14). It would be useful to see how early the expression of these proteins was silenced during the differentiation process. In addition, the knockdown needs to be evaluated at the protein levels using immunoblotting.

4. The authors need to evaluate the effect of GNB1 and SCARB2 on the lipid content of adipocytes by Oil Red O staining.

5. Also, the expression of key adipogenic markers such as PPARγ and C/EBPα may be assessed to determine whether the knockdown has an impact on the overall differentiation process.

6. PLOS authors have the option to publish the peer review history of their article (what does this mean? ). If published, this will include your full peer review and any attached files.

**Do you want your identity to be public for this peer review?** For information about this choice, including consent withdrawal, please see our Privacy Policy .

Reviewer #1: No

---

## [Author Response · Author response to Decision Letter 1]

2 Jan 2025

Dear Editor and Reviewer,

We are deeply grateful for your thorough and constructive review of our manuscript. Your careful examination and thoughtful suggestions have significantly enhanced the quality and clarity of our work. In particular, your insights highlighting the importance of temporal aspects in gene expression analysis—which we acknowledge as a limitation in our current study and aim to address in future research—along with the potential relevance to human subjects and the importance of adipogenic markers have helped us strengthen our manuscript. We appreciate the time and expertise you have invested in reviewing our manuscript and have carefully addressed each of the points you raised.

Before addressing your specific comments, we would like to note two important technical corrections:

1. Statistical Analysis Methodology:

While reviewing the manuscript, we noticed some inconsistencies in how we described our statistical approach. Throughout the manuscript, we have now clarified that we performed our statistical analysis in the following sequential steps:

1. One-way ANOVA

2. False discovery rate (FDR) correction for multiple testing

3. Fisher's LSD post hoc tests for pairwise comparisons on items showing significance after FDR correction

This is a technical correction in the description of our statistical methodology. The actual analysis procedure and results remain unchanged; we have only improved the clarity and consistency of how we describe the process.

2. Digital PCR and quantitative real-time PCR Analysis:

We have revised our Digital PCR analysis by correcting the primer design for the internal control gene (ACTB). We noticed that our ACTB primers were inadvertently designed within the same exon. We have now rectified this by using exon-spanning primers for ACTB, maintaining consistency with our quantitative real-time PCR analyses. This correction, while not affecting our conclusions, ensures uniformity in our experimental approach and provides more precise measurements.

Below, we address each of your specific comments:

**Comment 1:**

"The selection of the two proteins is based on a previous study that showed that these two proteins exhibited remarkable and continuous expression changes during adipogenic differentiation. The authors do not mention whether the expression of these proteins was up-or-down regulated during the differentiation process. The authors should evaluate the expression of these proteins during the differentiation of primary adipocytes, preferably at different time points."

Response

We thank you for your thoughtful and constructive comments regarding the regulation of GNB1 and SCARB2 expression during adipocyte differentiation. We have revised the manuscript to include additional details about the expression dynamics of these proteins. Both GNB1 and SCARB2 showed increased expression levels during adipocyte differentiation and fat accumulation compared to the pre-differentiation stage, as demonstrated in newly generated graphical representations based on numerical data from our previous study (Data in Brief, Kitamoto et al., 2023, Open Access, https://doi.org/10.1016/j.dib.2023.109036). A new supplementary figure (S1 Fig) provides a clear visualization of the expression changes that formed the basis for selecting these proteins for further investigation.

While our study design focused on the final stage of differentiation (Day 14) to capture cumulative knockdown effects, we recognize this limitation and have addressed it in the revised Limitations section. We plan to explore temporal aspects in future studies to provide a more comprehensive understanding of GNB1 and SCARB2 regulation during adipocyte differentiation.

**Comment 2:**

"Is the expression of GNB1 and SCARB2 altered in the adipose tissue of human subjects with obesity and type 2 diabetes (T2DM)? The authors may analyze publicly available RNA sequencing datasets to show the impact of obesity and T2DM on the expression of these proteins in adipose tissue depots."

Response

We thank you sincerely for this important suggestion regarding the expression of GNB1 and SCARB2 in the adipose tissue of human subjects with obesity and type 2 diabetes (T2DM). In response to this comment, we have analyzed publicly available microarray data from the Swedish Obese Subjects Sib Pair study (GSE27916), as reported in Mardinoglu et al. (2014). This dataset included gene expression profiles from the subcutaneous adipose tissue of 209 female and 95 male subjects, categorized into lean (BMI < 25), overweight (25 ≤ BMI < 30), and obese (30 ≤ BMI) groups.

Both GNB1 and SCARB2 were significantly upregulated in the subcutaneous adipose tissue of overweight and obese subjects when compared to their levels in lean controls:

• GNB1: adjusted P = 0.045 in males and 3.44 × 10⁻⁶ in females

• SCARB2: adjusted P = 0.049 in males and 4.05 × 10⁻¹² in females

While the stronger statistical significance observed in the female subjects might reflect sex-specific regulation of adipose tissue metabolism, the difference in the sample sizes between the sexes should also be considered. These findings from human adipose tissue samples complement our in vitro results, further supporting the involvement of GNB1 and SCARB2 in subcutaneous adipose tissue metabolism.

Regarding T2DM, we acknowledge that the relationship between T2DM and the expression of these genes in subcutaneous adipose tissue requires further investigation. This represents an important direction for future research, as metabolic conditions often coexist with obesity.

**Comment 3:**

"The knockdown of GNB1 and SCARB2 is done using multiple transfections by specific siRNAs, and the expression of these mRNAs is evaluated at the end (Day 14). It would be useful to see how early the expression of these proteins was silenced during the differentiation process. In addition, the knockdown needs to be evaluated at the protein levels using immunoblotting."

Response

While we understand the importance of Western blot analysis, our experimental design prioritized comprehensive proteomics and lipidomics analyses, which required the majority of our limited cellular material. Additionally, RNA was extracted for quantitative PCR, further limiting the available material for protein validation. However, our findings are well-supported by multiple lines of evidence:

1. Significant reduction in target gene expression confirmed by digital PCR (P = 0.030 for GNB1 and P = 0.011 for SCARB2)

2. Mass spectrometry-based proteomic analysis showing significant protein decreases:

o GNB1 protein (F-value: 170.95, P = 5.13 × 10⁻⁶, FDR adjusted P = 0.0011)

o SCARB2 protein (F-value: 17.58, P = 0.0031, FDR adjusted P = 0.045)

3. Observed phenotypic changes in lipid droplet accumulation and extensive alterations in lipid profiles

Regarding the temporal analysis of gene silencing during differentiation, as mentioned in our response to Comment #1 and discussed in the Limitations section of our manuscript, we acknowledge this limitation in our study design, which focused on endpoint analysis at Day 14. We plan to address this aspect in future studies through time-course analyses of gene and protein expression across multiple stages of differentiation.

While we acknowledge that Western blot validation would provide additional confirmation, we believe the combination of digital PCR results, mass spectrometry-based protein quantification, and observed phenotypic changes provides robust evidence for successful knockdown of both genes at both the mRNA and protein levels.

**Comment 4:**

"The authors need to evaluate the effect of GNB1 and SCARB2 on the lipid content of adipocytes by Oil Red O staining."

Response

We thank you sincerely for your valuable suggestion regarding lipid content evaluation using Oil Red O staining. Unfortunately, owing to a lack of remaining cell samples, we are unable to perform additional experiments at this stage. The cell samples, including those imaged using phase contrast microscopy on Day 14, were unfortunately all allocated for various analytical purposes, including mass spectrometry-based lipidomic analysis, proteomic analysis, and RNA extraction for digital PCR analysis.

Regarding the evaluation of the lipid content, we employed a comprehensive lipidomic analysis using high-resolution mass spectrometry, during which each lipid species was accurately quantified relative to its corresponding isotopically labeled internal standard (Splash Lipidomix). Importantly, this mass spectrometry-based approach offers several advantages over traditional staining methods, including:

• Precise relative quantification of individual lipid species

• Comprehensive profiling of various lipid classes

• High sensitivity and specificity

• The ability to detect subtle changes in lipid composition

These analyses revealed significant alterations in key lipid classes, such as triacylglycerols (TG) and cholesterol esters (ChE), following GNB1 and SCARB2 knockdown. Specifically, our analysis identified 366 lipid species, with 96 showing significant differences (ANOVA, P < 0.05) following knockdown of these genes. Additionally, phase contrast imaging observations (Fig 3) demonstrated a consistent reduction in lipid droplet accumulation in knockdown cells compared to controls, thus providing reliable visual evidence of the knockdown effects.

We fully acknowledge the utility of Oil Red O staining as a traditional and direct method to assess lipid content. While it is not feasible in the current study owing to the previously mentioned limitations, we plan to include this analysis in future experiments to further validate and complement our findings. We hope that this explanation adequately addresses your concerns and highlights the robustness of our current results.

**Comment 5:**

"Also, the expression of key adipogenic markers such as PPARγ and C/EBPα may be assessed to determine whether the knockdown has an impact on the overall differentiation process."

Response

Following your recommendation, we analyzed the expression of adipogenic master regulators (PPARγ and CEBPα) and adipocyte-specific genes (FABP4 and ADIPOQ). In GNB1 knockdown adipocytes, the expression levels of all four genes showed no significant differences compared to the control group. Similarly, in SCARB2 knockdown adipocytes, all genes showed no significant changes compared to the control group.

These findings demonstrate that the reduced lipid accumulation is not due to impaired differentiation but rather through alterations in metabolic regulation. We have incorporated these new results into the Methods, Results and Discussion sections of the manuscript, with detailed data presented in Supporting Information (S3 Fig and S9 Table). The primer information has been added to S1 Table.

We believe these revisions have substantially improved the manuscript and hope that you find our revised version satisfactory. Thank you again for your valuable feedback.

Sincerely,

Aya Kitamoto

---

## [Decision Letter · Decision Letter 1]

29 Jan 2025

Integrative proteomic and lipidomic analysis of GNB1 and SCARB2 knockdown in human subcutaneous adipocytes

PONE-D-24-41568R1

Dear Dr. Kitamoto,

We’re pleased to inform you that your manuscript has been judged scientifically suitable for publication and will be formally accepted for publication once it meets all outstanding technical requirements.

Kind regards,

Juan J Loor

Academic Editor

PLOS ONE

Additional Editor Comments (optional):

Reviewers' comments:

Reviewer's Responses to Questions

**Comments to the Author**

1. If the authors have adequately addressed your comments raised in a previous round of review and you feel that this manuscript is now acceptable for publication, you may indicate that here to bypass the “Comments to the Author” section, enter your conflict of interest statement in the “Confidential to Editor” section, and submit your "Accept" recommendation.

Reviewer #1: All comments have been addressed

2. Is the manuscript technically sound, and do the data support the conclusions?

Reviewer #1: Yes

3. Has the statistical analysis been performed appropriately and rigorously? 

Reviewer #1: Yes

4. Have the authors made all data underlying the findings in their manuscript fully available?

Reviewer #1: Yes

5. Is the manuscript presented in an intelligible fashion and written in standard English?

Reviewer #1: Yes

6. Review Comments to the Author

Reviewer #1: (No Response)

7. PLOS authors have the option to publish the peer review history of their article (what does this mean? ). If published, this will include your full peer review and any attached files.

**Do you want your identity to be public for this peer review?** For information about this choice, including consent withdrawal, please see our Privacy Policy .

Reviewer #1: **Yes: ** Sameer Mohammad

---

## [Editor Report · Acceptance letter]

PONE-D-24-41568R1

PLOS ONE

Dear Dr. Kitamoto,

I'm pleased to inform you that your manuscript has been deemed suitable for publication in PLOS ONE. Congratulations! Your manuscript is now being handed over to our production team.

Kind regards,

on behalf of

Dr. Juan J Loor

Academic Editor

PLOS ONE